# PSEUDO NUMERICAL METHODS FOR DIFFUSION MODELS ON MANIFOLDS

**Luping Liu, Yi Ren, Zhijie Lin & Zhou Zhao**[*]
Zhejiang University
{luping.liu,rayeren,linzhijie,zhaozhou}@zju.edu.cn

## ABSTRACT

Denoising Diffusion Probabilistic Models (DDPMs) can generate high-quality samples such as image and audio samples. However, DDPMs require hundreds to thousands of iterations to produce final samples. Several prior works have successfully accelerated DDPMs through adjusting the variance schedule (e.g., Improved Denoising Diffusion Probabilistic Models) or the denoising equation (e.g., Denoising Diffusion Implicit Models (DDIMs)). However, these acceleration methods cannot maintain the quality of samples and even introduce new noise at a high speedup rate, which limit their practicability. To accelerate the inference process while keeping the sample quality, we provide a fresh perspective that DDPMs should be treated as solving differential equations on manifolds. Under such a perspective, we propose pseudo numerical methods for diffusion models (PNDMs). Specifically, we figure out how to solve differential equations on manifolds and show that DDIMs are simple cases of pseudo numerical methods. We change several classical numerical methods to corresponding pseudo numerical methods and find that the pseudo linear multi-step method is the best in most situations. According to our experiments, by directly using pre-trained models on Cifar10, CelebA and LSUN, PNDMs can generate higher quality synthetic images with only 50 steps compared with 1000-step DDIMs (20x speedup), significantly outperform DDIMs with 250 steps (by around 0.4 in FID) and have good generalization on different variance schedules.[1]

## 1 INTRODUCTION

Denoising Diffusion Probabilistic Models (DDPMs) (Sohl-Dickstein et al., 2015; Ho et al., 2020) is a class of generative models which model the data distribution through an iterative denoising process reversing a multi-step noising process. DDPMs have been applied successfully to a variety of applications, including image generation (Ho et al., 2020; Song et al., 2020b), text generation (Hoogeboom et al., 2021; Austin et al., 2021), 3D point cloud generation (Luo & Hu, 2021), text-to-speech (Kong et al., 2021; Chen et al., 2020) and image super-resolution (Saharia et al., 2021).

Unlike Generative Adversarial Networks (GANs) (Goodfellow et al., 2014), which require careful hyperparameter tuning according to different model structures and datasets, DDPMs can use similar model structures and be trained by a simple denoising objective which makes the models fit the noise in the data. To generate samples, the iterative denoising process starts from white noise and progressively denoises it into the target domain according to the noise predicted by the model at every step. However, a critical drawback of DDPMs is that DDPMs require hundreds to thousands of iterations to produce high-quality samples and need to pass through a network at least once at every step, which makes the generation of a large number of samples extremely slow and infeasible. In contrast, GANs only need one pass through a network.

There have been many recent works focusing on improving the speed of the denoising process. Some works search for better variance schedules, including Nichol & Dhariwal (2021) and Watson et al. (2021). Some works focus on changing the inference equation, including Song et al. (2020a)

---

[*]Corresponding author
[1]Our implementation is available at https://github.com/luping-liu/PNDM.

and Song et al. (2020b). Denoising Diffusion Implicit Models (DDIMs) (Song et al., 2020a) relying on a non-Markovian process accelerate the denoising process by taking multiple steps every iteration. Probability Flows (PFs) (Song et al., 2020b) build a connection between the denoising process and solving ordinary differential equations and use numerical methods of differential equations to accelerate the denoising process. Additionally, we introduce more related works in Appendix A.1.

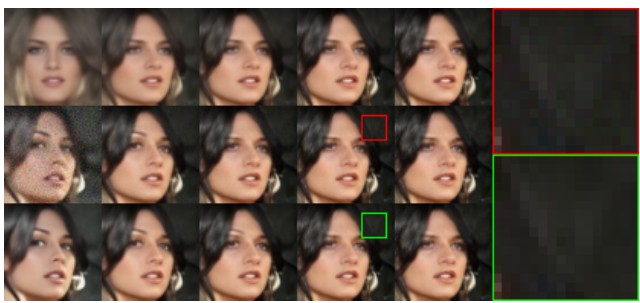

However, this direct connection between DDPMs and numerical methods (e.g., forward Euler method, linear multi-step method and Runge-Kutta method (Timothy, 2017)) has weaknesses in both speed and effect (see Section 3.1). Some numerical methods are straightforward, like the forward Euler method, but they can only trade quality for speed. Some numerical methods can accelerate the reverse process without loss of quality, like the Runge-Kutta method, but they need to propagate forward more times along a neural network at every step. Furthermore, we also notice that numerical methods can introduce noticeable noise at a high speedup rate, which makes high-order numerical methods (e.g., Runge-Kutta method) even less effective than DDIMs. This phenomenon is also mentioned in Salimans & Ho (2022).

Figure 1: 5, 10, 20, 50 and 100-steps generated results using DDIMs, classical numerical methods and PNDMs.

To figure out the reason for the performance degradation in classical numerical methods, we conduct some analyses and find that classical numerical methods may sample data far away from the main distribution area of the data, and the inference equations of DDPMs do not satisfy a necessary condition of numerical methods at the last several steps (see Section 3.2).

To tackle these problems, we design new numerical methods called pseudo numerical methods for diffusion models (PNDMs) to generate samples along a specific manifold in $\mathbb{R}^n$, which is the high-density region of the data. We first compute the corresponding differential equations of diffusion models directly and self-consistently, which builds a theoretical connection between DDPMs and numerical methods. Considering that classical numerical methods cannot guarantee to generate samples on certain manifolds, we provide brand-new numerical methods called pseudo numerical methods based on our theoretical analyses. We also find that DDIMs are simple cases of pseudo numerical methods, which means that we also provide a new way to understand DDIMs better. Furthermore, we find that the pseudo linear multi-step method is the fastest method for diffusion models under similar generated quality.

Besides, we provide a detailed theoretical analysis of our new theory and give visualization results to support our theory intuitively. According to our experiments, our methods have several advantages:

- Our methods combine the benefits of DDIMs and high-order numerical methods successfully. We theoretically prove that our new methods PNDMs are second-order convergent while DDIMs are first-order convergent, which makes PNDMs 20x faster without loss of quality on Cifar10 and CelebA.
- Our methods can reduce the best FID of pre-trained models with even shorter sampling time. With only 250 steps, our new denoising process can reduce the best FID by around 0.4 points Cifar10 and CelebA. We achieve a new SOTA FID score of 2.71 on CelebA.
- Our methods work well with different variance schedules, which means that our methods have a good generalization and can be used together with those works introducing better variance schedules to accelerate the denoising process further.

## 2 BACKGROUND

In this section, we introduce some backgrounds. Firstly, we present the classical understanding of DDPMs. Then we provide another understanding based on Song et al. (2020b), which inspires us to use numerical methods to accelerate the denoising process of diffusion models. After that, we introduce some background on numerical methods used later in this paper.

## 2.1 Denoising Diffusion Probabilistic Models

DDPMs model the data distribution from Gaussian distribution to image distribution through an iterative denoising process. Let $x_0$ be an image, then the diffusion process is a Markov process and the reverse process has a similar form to the diffusion process, which satisfies:

$$\begin{aligned} x_{t+1} &\sim \mathcal{N}(\sqrt{1-\beta_t}x_t, \beta_t\mathrm{I}), \ t = 0, 1, \cdots, N-1. \\ x_{t-1} &\sim \mathcal{N}(\mu_\theta(x_t, t), \beta_\theta(x_t, t)\mathrm{I}), \ t = N, N-1, \cdots, 1. \end{aligned} \tag{1}$$

Here, $\beta_t$ controls the speed of adding noise to the data, calling them the variance schedule. $N$ is the total number of steps of the denoising process. $\mu_\theta$ and $\beta_\theta$ are two neural networks, and $\theta$ are their parameters.

Ho et al. (2020) get some statistics estimations of $\mu_\theta$ and $\beta_\theta$. According to the properties of the conditional Gaussian distribution, we have:

$$\begin{aligned} q(x_t|x_0) &= \mathcal{N}(\sqrt{\bar{\alpha}_t}x_0, (1-\bar{\alpha}_t)\mathrm{I}), \\ q(x_{t-1}|x_t, x_0) &= \mathcal{N}(\bar{\mu}_t(x_t, x_0), \bar{\beta}_t\mathrm{I}). \end{aligned} \tag{2}$$

Here, $\alpha_t = 1 - \beta_t$, $\bar{\alpha}_t = \prod_{i=1}^t \alpha_i$, $\bar{\mu}_t = \frac{\sqrt{\bar{\alpha}_{t-1}}\beta_t}{1-\bar{\alpha}_t}x_0 + \frac{\sqrt{\alpha_t}(1-\bar{\alpha}_{t-1})}{1-\bar{\alpha}_t}x_t$ and $\bar{\beta}_t = \frac{1-\bar{\alpha}_{t-1}}{1-\bar{\alpha}_t}\beta_t$. Then this paper sets $\beta_\theta = \bar{\beta}_t$ and designs a objective function to help neural networks to represent $\mu_\theta$.

**Objective Function** The objective function is defined by:

$$\begin{aligned} L_{t-1} &= \mathbb{E}_q\left[||\bar{\mu}_t(x_t, x_0) - \mu_\theta(x_t, t)||^2\right] \\ &= \mathbb{E}_{x_0, \epsilon}\left[||\frac{1}{\sqrt{\alpha_t}}\left(x_t(x_0, \epsilon) - \frac{\beta_t}{\sqrt{1-\bar{\alpha}_t}}\epsilon\right) - \mu_\theta(x_t(x_0, \epsilon), t)||^2\right] \\ &= \mathbb{E}_{x_0, \epsilon}\left[\frac{\beta_t^2}{\alpha_t(1-\bar{\alpha}_t)}||\epsilon - \epsilon_\theta(\sqrt{\bar{\alpha}_t}x_0 + \sqrt{1-\bar{\alpha}_t}\epsilon, t)||^2\right]. \end{aligned} \tag{3}$$

Here, $x_t(x_0, \epsilon) = \sqrt{\bar{\alpha}_t}x_0 + \sqrt{1-\bar{\alpha}_t}\epsilon$, $\epsilon \sim \mathcal{N}(0, 1)$, $\epsilon_\theta$ is an estimate of the noise $\epsilon$. The relationship between $\mu_\theta$ and $\epsilon_\theta$ is $\mu_\theta = \frac{1}{\sqrt{\alpha_t}}(x_t - \frac{\beta_t}{\sqrt{1-\bar{\alpha}_t}}\epsilon_\theta)$. Because $\epsilon \sim \mathcal{N}(0, 1)$, we assume that the mean and variance of $\epsilon_\theta$ are 0 and 1.

## 2.2 Stochastic Differential Equation

According to Song et al. (2020b), there is another understanding of DDPMs. The diffusion process can be treated as solving a certain stochastic differential equation $dx = (\sqrt{1-\beta(t)} - 1)x(t)dt + \sqrt{\beta(t)}dw$. According to Anderson (1982), the denoising process also satisfies a similar stochastic differential equation:

$$dx = \left((\sqrt{1-\beta(t)} - 1)x(t) - \beta(t)\epsilon_\theta(x(t), t)\right)dt + \sqrt{\beta(t)}d\bar{w}. \tag{4}$$

This is Variance Preserving stochastic differential equations (VP-SDEs). Here, we change the domain of $t$ from $[1, N]$ to $[0, 1]$. When $N$ tends to infinity, $\{\beta_i\}_{i=1}^N$, $\{x_i\}_{i=1}^N$ become continuous functions $\beta(t)$ and $x(t)$ on $[0, 1]$. Song et al. (2020b) also show that this equation has an ordinary differential equation (ODE) version with the same marginal probability density as Equation (4):

$$dx = \left((\sqrt{1-\beta(t)} - 1)x(t) - \frac{1}{2}\beta(t)\epsilon_\theta(x(t), t)\right)dt. \tag{5}$$

This different denoising equation with no random item and the same diffusion equation together is Probability Flows (PFs). These two denoising equations show us a new possibility that we can use numerical methods to accelerate the reverse process. As far as we know, DDIMs first try to remove this random item, so PFs can also be treated as an acceleration of DDIMs, while VP-SDEs are an acceleration of DDPMs.

## 2.3 Numerical Method

Many classical numerical methods can be used to solve ODEs, including the forward Euler method, Runge-Kutta method and linear multi-step method (Timothy, 2017).

**Forward Euler Method** For a certain differential equation satisfying $\frac{dx}{dt} = f(x,t)$. The trivial numerical method is forward Euler method satisfying $x_{t+\delta} = x_t + \delta f(x_t, t)$.

**Runge-Kutta Method** Runge-Kutta method uses more information at every step, so it can achieve higher accuracy [2]. Runge-Kutta method satisfies:

$$\begin{cases} k_1 = f(x_t, t) & , \quad k_2 = f(x_t + \frac{\delta}{2}k_1, t + \frac{\delta}{2}) \\ k_3 = f(x_t + \frac{\delta}{2}k_2, t + \frac{\delta}{2}), & k_4 = f(x_t + \delta k_3, t + \delta) \\ x_{t+\delta} = x_t + \frac{\delta}{6}(k_1 + 2k_2 + 2k_3 + k_4). \end{cases} \quad (6)$$

**Linear Multi-Step Method** Linear multi-step method is another numerical method and satisfies:

$$x_{t+\delta} = x_t + \frac{\delta}{24}(55f_t - 59f_{t-\delta} + 37f_{t-2\delta} - 9f_{t-3\delta}), \ f_t = f(x_t, t). \quad (7)$$

## 3 PSEUDO NUMERICAL METHOD FOR DDPM

In this section, we first compute the corresponding differential equations of diffusion models to build a direct connection between DDPMs and numerical methods. As a byproduct, we can directly use pre-trained models from DDPMs. After establishing this connection, we provide detailed analyses on the weakness of classical numerical methods. To solve the problems in classical numerical methods, we dive into the structure of numerical methods by dividing their equations into a gradient part and a transfer part and define pseudo numerical methods by introducing nonlinear transfer parts. We find that DDIMs can be regarded as simple pseudo numerical methods. Then, We explore the pros and cons of different numerical methods and choose the linear multi-step method to make numerical methods faster. Finally, we summarize our findings and analyses and safely propose our novel pseudo numerical methods for diffusion models (PNDMs), which combine our proposed transfer part and the gradient part of the linear multi-step method. Furthermore, we analyze the convergence order of pseudo numerical methods to demonstrate the effectiveness of our methods theoretically.

### 3.1 FORMULA TRANSFORMATION

According to Song et al. (2020a), the reverse process of DDPMs and DDIMs satisfies:

$$x_{t-1} = \sqrt{\bar{\alpha}_{t-1}} \left( \frac{x_t - \sqrt{1 - \bar{\alpha}_t}\epsilon_\theta(x_t, t)}{\sqrt{\bar{\alpha}_t}} \right) + \sqrt{1 - \bar{\alpha}_{t-1} - \sigma_t^2}\epsilon_\theta(x_t, t) + \sigma_t \epsilon_t. \quad (8)$$

Here, $\sigma_t$ controls the ratio of random noise. If $\sigma_t$ equals one, Equation (8) represents the reverse process of DDPMs; if $\sigma_t$ equals zero, this equation represents the reverse process of DDIMs. And only when $\sigma_t$ equals zero, this equation removes the random item and becomes a discrete form of a certain ODE. Theoretically, the numerical methods that can be used on differential equations with random items are limited. And Song et al. (2020b) have done enough research in this case. Empirically, Song et al. (2020a) have shown that DDIMs have a better acceleration effect when the number of total steps is relatively small. Therefore, our work concentrate on the case $\sigma_t$ equals zero.

To find the corresponding ODE of Equation (8), we replace discrete $t - 1$ with a continuous version $t - \delta$ according to (Song et al., 2020a) and change this equation into a differential form, namely, subtract $x_t$ from both sides of this equation:

$$x_{t-\delta} - x_t = (\bar{\alpha}_{t-\delta} - \bar{\alpha}_t) \left( \frac{x_t}{\sqrt{\bar{\alpha}_t}(\sqrt{\bar{\alpha}_{t-\delta}} + \sqrt{\bar{\alpha}_t})} - \frac{\epsilon_\theta(x_t, t)}{\sqrt{\bar{\alpha}_t}(\sqrt{(1 - \bar{\alpha}_{t-\delta})\bar{\alpha}_t} + \sqrt{(1 - \bar{\alpha}_t)\bar{\alpha}_{t-\delta}})} \right). \quad (9)$$

Because $\delta$ is a continuous variable from 0 to $t$, we can now compute the derivative of the generation data $x_t$ and get that $\lim_{\delta \to 0} \frac{x_t - x_{t-\delta}}{\delta} = -\bar{\alpha}'(t) \left( \frac{x(t)}{2\bar{\alpha}(t)} - \frac{\epsilon_\theta(x(t), t)}{2\bar{\alpha}(t)\sqrt{1 - \bar{\alpha}(t)}} \right)$. Here, $\bar{\alpha}(t)$ is the continuous version of $\{\bar{\alpha}_i\}_{i=1}^N$ like the definition of $x(t)$. Therefore, the corresponding ODE when $\delta$ tends to

---

[2]To achieve higher accuracy, more information is not enough. The reason why these methods achieve higher accuracy can be found in Appendix A.2

zero of Equation (9) is:

$$\frac{dx}{dt} = -\bar{\alpha}'(t) \left( \frac{x(t)}{2\bar{\alpha}(t)} - \frac{\epsilon_\theta(x(t), t)}{2\bar{\alpha}(t)\sqrt{1 - \bar{\alpha}(t)}} \right). \tag{10}$$

## 3.2 CLASSICAL NUMERICAL METHOD

After getting the target ODE, the easiest way to solve it is through classical numerical methods. However, We notice that classical numerical methods can introduce noticeable noise at a high speedup rate, making high-order numerical methods (e.g., Runge-Kutta method) even less effective than DDIMs. This phenomenon is also mentioned in Salimans & Ho (2022). To make better use of numerical methods, we analyze the differences between Equation (10) and usual differential equations and find two main problems when we directly use numerical methods with diffusion models.

The first problem is that the neural network $\epsilon_\theta$ and Equation (10) are well-defined only in a limited area. Equation (2) shows that the data $x_t$ is generated along a curve close to an arc. According to Figure 2, most of $x_t$ is concentrated in a band with a width of around 0.1, namely the red area in Figure 2. This means that the neural network $\epsilon_\theta$ cannot get enough examples to fit the noise successfully away from this area. Therefore, $\epsilon_\theta$ and Equation (10), which contains $\epsilon_\theta$, are only well-defined in this limited area. However, all classical numerical methods generate results along a straight line instead of an arc. The generation process may generate samples away from the well-defined area and then introduce new errors. In Section 4.3 we will give more visualization results to support this.

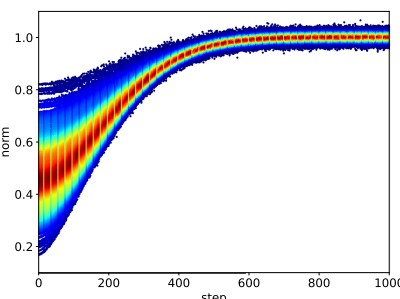

Figure 2: the density distribution of the norm of the data.

The second problem is that Equation (10) is unbounded at most cases. We find that for most linear variance schedules $\beta_t$, Equation (10) tends to infinity when $t$ tends to zero (see Appendix A.4), which does not satisfy the condition of numerical methods mentioned in Section 2.3. This is an apparent theoretical weakness that previous works have not explored. On the contrary, in the original DDPMs and DDIMs, the prediction of the sample $x_t$ and the noise $\epsilon_\theta$ in the data are more and more precise as the index $t$ tends to zero (see Appendix A.5). This means original diffusion models do not make a significant error in the last several steps, whereas using numerical methods on Equation (10) does. This explains why DDIMs are better than higher-order numerical methods.

## 3.3 PSEUDO NUMERICAL METHOD ON MANIFOLD

The first problem above shows that we should try to solve our problems on certain manifolds. Here, target manifolds are the high-density region of the data $x_t$ of DDPMs, which is defined by $x_t(x_0, \epsilon) = \sqrt{\bar{\alpha}_t} x_0 + \sqrt{1 - \bar{\alpha}_t} \epsilon, \epsilon \sim \mathcal{N}(0, 1)$. Ernst & Gerhard (1996) show several numerical methods to solve differential equations on manifolds that have analytic expressions. Unfortunately, it's challenging to use the above expression of manifolds. Because we do not know the target $x_0$ in the reverse process and random items $\epsilon$ are hard to handle, too.

In this paper, we design a different way that we make our new equation of denoising process more fits with the equation of original DDIMs to make their results share similar data distribution. Firstly, we divide the classical numerical methods into two parts: gradient and transfer parts. The gradient part determines the gradient at each step, while the transfer part generates the result at the next step. For example, linear multi-step method can be divided into the gradient part $f' = \frac{\delta}{24}(55 f_t - 59 f_{t-\delta} + 37 f_{t-2\delta} - 9 f_{t-3\delta})$ and the transfer part $x_{t+\delta} = x_t + \delta f'$. All classical numerical methods have the same linear transfer part, while gradient parts are different.

We define those numerical methods which use a nonlinear transfer part as pseudo numerical methods. And an expected transfer part should have the property that when the result from the gradient part is precise, then the result of the transfer part is as close to the manifold as possible and the error of this result is as small as possible. We find that Equation (9) satisfies such property.

**Property 3.1** *If $\epsilon$ is the precise noise in $x_t$, then the result of $x_{t-\delta}$ from Equation (9) is also precise.*

And we put the proof of this property in Appendix A.5. Therefore, we use:

$$\phi(x_t, \epsilon_t, t, t-\delta) = \frac{\sqrt{\bar{\alpha}_{t-\delta}}}{\sqrt{\bar{\alpha}_t}} x_t - \frac{(\bar{\alpha}_{t-\delta} - \bar{\alpha}_t)}{\sqrt{\bar{\alpha}_t}(\sqrt{(1-\bar{\alpha}_{t-\delta})\bar{\alpha}_t} + \sqrt{(1-\bar{\alpha}_t)\bar{\alpha}_{t-\delta}})} \epsilon_t \qquad (11)$$

as the transfer part and $\epsilon_\theta$ as the gradient part. That if $\epsilon_\theta$ is precise, the result of $x_{t-\delta}$ is also precise, which means that $\epsilon_\theta$ can determine the direction of the denoising process to generate the final results. Therefore, such a choice also satisfies the definition of a gradient part. Now, we have our gradient part $\epsilon_\theta$ and transfer part $\phi$.

This combination solves the two problems mentioned above successfully. Firstly, our new transfer parts do not introduce new errors. This property also means that it keeps the results at the next step on the target manifold because generating samples away is a kind of error. This shows that we solve the first problem. Secondly, we know that the prediction of $\epsilon_\theta$ is more and more precise in the reverse process in the above subsection. And our new transfer part can generate precise results according to the precise prediction of $\epsilon_\theta$. Therefore, our generation results are more and more precise using pseudo numerical methods, while classical numerical methods can introduce obvious error at the last several steps. This shows that we solve the second problem, too. We also find that their combination $\phi(x_t, \epsilon_\theta(x_t, t), t, t-1)$ is just the inference equation used by DDIMs, so DDIMs is a simple case of pseudo numerical methods. Here, we define DDIMs as DDIMs*, emphasizing that it is a pseudo numerical method.

### 3.4 GRADIENT PART

Because we split numerical methods into two parts, we can use the same gradient part from different classical numerical methods freely (e.g., linear multi-step method), although we change the transfer part of our inference equation. Our theoretical analyses and experiments show that the gradient part from different classical methods can work well with our new transfer part (see Section 3.6, 4.2). By using the same gradient part of the linear multi-step method, we have:

$$\begin{cases} e_t = \epsilon_\theta(x_t, t) \\ e'_t = \frac{1}{24}(55e_t - 59e_{t-\delta} + 37e_{t-2\delta} - 9e_{t-3\delta}) \\ x_{t+\delta} = \phi(x_t, e'_t, t, t+\delta). \end{cases} \qquad (12)$$

By using the same gradient part of Runge-Kutta method, we have:

$$\begin{cases} e^1_t = \epsilon_\theta(x_t, t) \\ x^1_t = \phi(x_t, e^1_t, t, t+\frac{\delta}{2}) \\ e^2_t = \epsilon_\theta(x^1_t, t+\frac{\delta}{2}) \\ x^2_t = \phi(x_t, e^2_t, t, t+\frac{\delta}{2}) \\ e^3_t = \epsilon_\theta(x^2_t, t+\frac{\delta}{2}) \\ x^3_t = \phi(x_t, e^3_t, t, t+\delta) \\ e^4_t = \epsilon_\theta(x^4_t, t+\delta) \\ e'_t = \frac{1}{6}(e^1_t + 2e^2_t + 2e^3_t + e^4_t) \\ x_{t-\delta} = \phi(x_t, e'_t, t, t+\delta). \end{cases} \qquad (13)$$

---

**Algorithm 1** DDIMs

1: $x_T \sim \mathcal{N}(0, I)$
2: **for** $t = T-1, \cdots, 1, 0$ **do**
3: $\quad x_t = \phi(x_{t+1}, \epsilon_\theta(x_{t+1}, t+1), t+1, t)$
4: **end for**
5: **return** $x_0$

---

**Algorithm 2** PNDMs

1: $x_T \sim \mathcal{N}(0, I)$
2: **for** $t = T-1, T-2, T-3$ **do**
3: $\quad x_t, e_t = \text{PRK}(x_{t+1}, t+1, t)$
4: **end for**
5: **for** $t = T-4, \cdots, 1, 0$ **do**
6: $\quad x_t, e_t = \text{PLMS}(x_{t+1}, \{e_p\}_{p>t}, t+1, t)$
7: **end for**
8: **return** $x_0$

---

Abbreviate Equation (12) and (13) as $x_{t+\delta}, e_t = \text{PLMS}(x_t, \{e_p\}_{p<t}, t, t+\delta)$, $x_{t+\delta}, e^1_t = \text{PRK}(x_t, t, t+\delta)$.

Here, we have provided three kinds of pseudo numerical methods. Although advanced numerical methods can accelerate the denoising process, some may have to compute the gradient part $\epsilon_\theta$ more times at every step, like the Runge-Kutta method. Propagating forward four times along a neural network makes the denoising process slower. However, we find that the linear multi-step method can reuse the result of $\epsilon_\theta$ four times and only compute $\epsilon_\theta$ once at every step. And theoretical analyses tell us that the Runge-Kutta and linear multi-step method have the same convergence order and similar results.

Therefore, we use the gradient part of the linear multi-step method and our new transfer part as our main pseudo numerical methods for diffusion models (PNDMs). In Table 1, we show the relationship between different numerical methods. Here, we can see PNDMs combine the benefits of higher-order classical numerical methods (in the gradient part) and DDIMs (in the transfer part).

| order $\phi$ | first | non-first |
|---|---|---|
| linear | forward Euler | linear multi-step, Runge-Kutta... |
| nonlinear | DDIM | PNDM |

Table 1: The relationship between different numerical methods.

### 3.5 ALGORITHM

We can provide our whole algorithm of the denoising process of DDIMs now. According to Song et al. (2020a), the algorithm of the original method satisfies Algorithm 1. And our new algorithm of DNPMs uses the pseudo linear multi-step and pseudo Runge-Kutta method, which satisfies Algorithm 2. Here, we cannot use linear multi-step initially because the linear multi-step method cannot start automatically, which needs at least three previous steps' information to generate results. So we use the Runge-Kutta method to compute the first three steps' results and then use the linear multi-step method to calculate the remaining.

We also use the gradient parts of two second-order numerical methods to get another pseudo numerical method. We introduce the details of this method in Appendix A.3. We call it S-PNDMs, because its gradient part uses information from two steps at every step. Similarly, we also call our first PNDMs F-PNDMs, which use data from four steps, when we need to distinguish them.

### 3.6 CONVERGENCE ORDER

Change the transfer part of numerical methods may introduce unknown error. To determine the influence of our new transfer part theoretically, we compute the local and global error between the theoretical result of Equation (10) $x(t+\delta)$ and our new methods, we find that $x(t+\delta) - x_{\text{DDIM}}(x+\delta) = O(\delta^2)$ and

$$x(t+\delta) - x_{\text{S/F-PNDM}}(x+\delta) = O(\delta^3). \tag{14}$$

If the target ODE satisfies Lipschitz condition and local error $e_{\text{local}} = O(\delta^k)$, then there are $C$ and $h$ such that the global error $e_{\text{global}}$ satisfies $e_{\text{global}} \leq C\delta^k(1 + e^h + e^{2h} + \cdots) \leq C'\delta^{k-1}$. And we have that the convergence order is equal to the order of the global error. The detailed proof can be found in Appendix A.6. Therefore, we get the following property:

**Property 3.2** *S/F-PNDMs have third-order local error and are second-order convergent.*

## 4 EXPERIMENT

### 4.1 SETUP

We conduct unconditional image generation experiments on four datasets: Cifar10 ($32 \times 32$) (Krizhevsky et al., 2009), CelebA ($64 \times 64$) (Liu et al., 2015), LSUN-church ($256 \times 256$) and LSUN-bedroom ($256 \times 256$) (Yu et al., 2016). According to the analysis in Section 3.1, we can use pre-trained models from prior works in our experiments. The pre-trained models for Cifar10, LSUN-church and LSUN-bedroom are taken from Ho et al. (2020) and the pre-trained model for CelebA is taken from Song et al. (2020a). In these models, the number of total steps N is 1000 and the variance schedule is linear variance schedule. And we also use a pre-trained model for Cifar10, which uses a cosine variance schedule from improved denoising diffusion probabilistic models (iDDPMs (Nichol & Dhariwal, 2021)).

### 4.2 SAMPLE EFFICIENCY AND QUALITY

To analyze the acceleration effect, we test Fenchel Inception Distance (FID (Heusel et al., 2018)) on different datasets under different steps and different numerical methods, including DDIMs, S-PNDMs, F-PNDMs and classical fourth-order numerical methods (FONs) (e.g., Runge-Kutta

Table 2: Image generation measured in FID on Cifar10 and CelebA. PFs use black box ODE solvers and we use the number of score function evaluations as the step of PFs. DDIM* is a retest of DDIM. The bold results mean the best ones using the same pretrained model. We use the 50-step, 512 batch size experiment on an RTX-3090 to test the computational cost and the column time is the average computational cost per step in seconds. And we put the results of standard deviation in Appendix A.12

| dataset | FID \ step
model | 10 | 20 | 50 | 100 | 250 | 1000 | time |
|---|---|---|---|---|---|---|---|---|
| Cifar10 | DDIM | 13.4 | 6.84 | 4.67 | 4.16 | | 4.04 | |
| | PF | | 13.8 | 3.89 | 3.69 | 3.71 | 3.72 | |
| Cifar10
(linear) | DDIM* | 18.5 | 10.9 | 6.99 | 5.52 | 4.52 | 4.00 | 0.337 |
| | FON | 13.1 | 7.41 | 5.26 | 4.65 | 4.12 | 3.71 | 0.390 |
| | S-PNDM | 11.6 | 7.56 | 5.18 | 4.34 | 3.91 | 3.80 | 0.344 |
| | F-PNDM | 7.03 | 5.00 | 3.95 | 3.72 | **3.60** | 3.70 | 0.391 |
| Cifar10
(cosine) | DDIM | 14.5 | 8.79 | 5.86 | 4.92 | 4.30 | 3.69 | 0.505 |
| | S-PNDM | 8.64 | 5.77 | 4.46 | 3.94 | 3.71 | 3.38 | 0.517 |
| | F-PNDM | 7.05 | 4.61 | 3.68 | 3.53 | 3.49 | **3.26** | 0.595 |
| CelebA | DDIM | 17.3 | 13.7 | 9.17 | 6.53 | | 3.51 | |
| CelebA
(linear) | DDIM* | 16.9 | 13.4 | 8.95 | 6.36 | 4.44 | 3.41 | 1.237 |
| | FON | 16.0 | 11.6 | 8.13 | 6.70 | 5.14 | 4.17 | 1.431 |
| | S-PNDM | 12.2 | 9.45 | 5.69 | 4.03 | 3.19 | 2.99 | 1.258 |
| | F-PNDM | 7.71 | 5.51 | 3.34 | 2.81 | **2.71** | 2.86 | 1.433 |

method and linear multi-step method). On Cifar10 and CelebA, we first provide the results of previous works DDIMs. Then, we use the same pre-trained models to test numerical methods mentioned in this paper and put the results in Cifar10 / CelebA (linear). We also use models from iDDPMs to test nonlinear variance schedules and put the results in Cifar10 (cosine). Song et al. (2020b) do not provide detailed FID results of probability flows (PFs) under different steps, so we retest the results using its pretrained models by ourselves.

**Efficiency** Our two baselines are DDIM and PF. DDIM is a simple case of pseudo numerical methods, and PF is a case of classical numerical methods. However, PF uses a much bigger model than DDIM and uses some tricks to improve the sample quality. To ensure the experiment's fairness, we use fourth-order numerical methods on Equation (10) and the model from DDIM. In Table 2, we find that the performance of FON is limited when the number of steps is small. By contrast, our new methods, including S-PNDM and F-PNDM, can improve the generated results regardless of whether the number of steps used is large or small. According to Cifar10 / CelebA (linear), F-PNDM can achieve lower FID than 1000 steps DDIM using only 50 steps, making diffusion models 20x faster without losing quality.

We draw a line chart of computation cost with FID according to the results of Cifar10 (linear) above in Figure 3. Because F-PNDM uses the pseudo Runge-Kutta method to generate the first three steps, it is slower than other methods at the first several steps. Therefore, S-PNDM can achieve the best FID initially, then F-PNDM becomes the best and the acceleration is significant.

**Quality** When the number of steps is relatively big, the results of FON become more and more similar to that of pseudo numerical methods. This is because all the methods are solving Equation (10), and their convergent results should be the same. However, pseudo numerical methods still work better using a large number of steps empirically. F-PNDM can improve the best FID around 0.4 using pretrained models and achieves a new SOTA FID score of 2.71

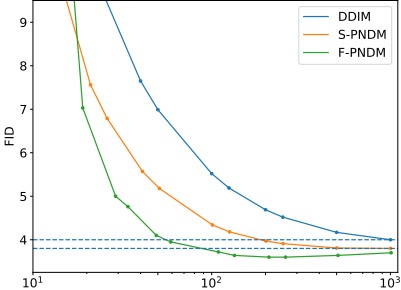

Figure 3: The FID results under different computation costs and different numerical methods on Cifar10. The unit of time is the computational cost of 1-step DDIM, which is 0.337s.

on CelebA, which shows that our work can not only accelerate diffusion models but also improve the sample quality topline. We also notice that the FID results of F-PNDM converge after more than 250 steps. The FID results will fluctuate around a value then. This phenomenon is more pronounced when we test our methods on LSUN (see Table 5, 6).

According to Cifar10 (cosine), the cosine variance schedule can lower FID using a relatively large number of steps. More analyses about variance schedule can be found in Appendix A.7. What's more, we test our methods on other datasets and provide our FID results in Appendix A.9 and image results in Appendix A.10. We can draw similar conclusions on our methods' acceleration and sampling quality, regardless of the datasets and the size of the images.

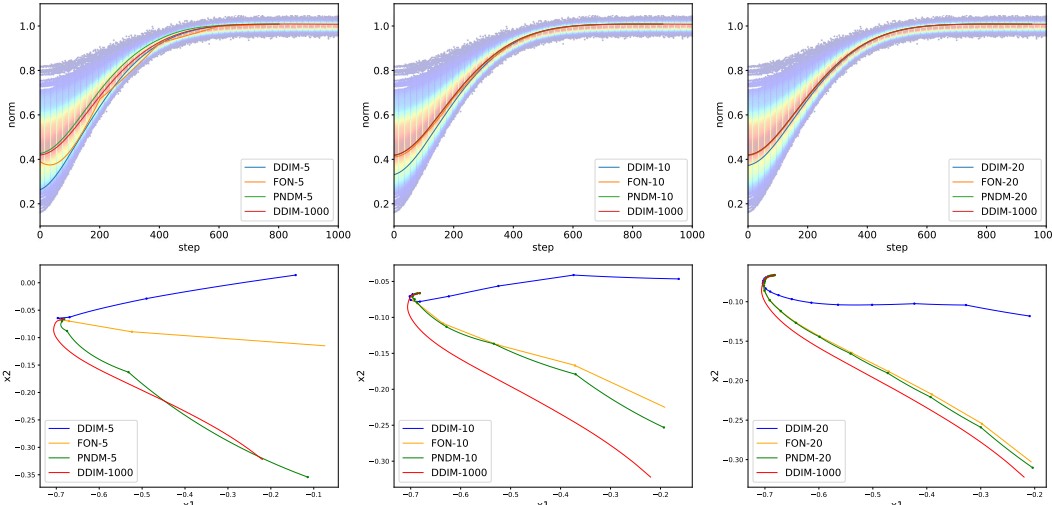

Figure 4: The upper part shows the change of norm with the number of steps using different methods and different steps. The lower part shows the generation curves of two points using different methods and different steps. DDIM-n means n-step DDIM method. Experiments in this subsection all use the Cifar10 dataset and we use the 1000-step DDIM's result as our target result.

## 4.3 SAMPLE ON MANIFOLDS

Here, we design visualization experiments to show the effort of our new methods and support our analyses. Because it is hard to visualize high-dimensional data, we use the change of a global characteristic norm and a local characteristic pixel to show the change of the data under different steps. For pixel, we randomly choose two positions $p^1, p^2$. Then for a series of images $x_T, x_{T-k}, \cdots, x_0$ derived from the reverse process, we denote $y_t^k$ as the value of $x_t$ at position $p^k$. Then we draw a polyline $(y_t^1, y_t^2)_{t=T,\cdots}$ in $\mathbb{R}^2$. For norm, we first count the distribution of the norm of the training datasets under different steps and use this to make a heat map as the background. After that, we draw the norm of our generated results using different methods and steps above this heat map.

In Figure 4, we can see that the FON may run far away from the high-density area of the data, which explains why FON may introduce noticeable noise. However, PNDM can avoid this problem and appropriately fit the target result. More visualization results supporting our analysis can be found in Appendix A.11. What's more, we design a toy example to test our new methods without the influence of neural networks and get similar conclusions as to the real cases above. We put the detailed results in Appendix A.8.

## 5 DISCUSSION

In this paper, we provided DNPMs, a new numerical method suitable for solving the corresponding ODEs of DDPMs. DNPMs can generate high-quality images using fewer steps without loss of quality successfully. Based on the idea of this work, further improvement can be explored in our future works: 1) find a better variance schedule for DNPMs: although we tested DNPMs on linear variance schedule and cosine variance schedule in this work, there might be another variance schedule more suitable for our proposed numerical methods. 2) Find higher-order convergent pseudo numerical methods: we analyzed the convergence order of S/F-DNPMs, which are both second-order convergent. However, F-DNPMs achieve better FID than S-DNPMs in most cases. We think this is because the result between our transfer part and target ODE has a higher-order error, which limits the convergence order of F-DNPMs. This error from the change of the transfer part is theoretical but does not influence the quality of images according to the property of Equation (11). However, making the transfer part higher-order convergent and finding the influence of such change is still exciting and needs more research. 3) Extend PNDMs to more general applications: when we proved the convergence order of DNPMs, we found that other kinds of transfer parts can keep the convergence order unchanged too, which means that pseudo numerical methods can be used on more general applications, like certain neural ODEs (Chen et al., 2019; Dupont et al., 2019).

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

## A APPENDIX

### A.1 RELATED WORK

DDPMs have been well developed in the last few years. Some works concentrate on improving the quality and the speed of DDPMs and making DDPMs more practical. Song et al. (2020a) introduce new inference equations to accelerate DDPMs. Nichol & Dhariwal (2021), Watson et al. (2021) and Kong & Ping (2021) choose to find better variance schedules to improve the images quality. Vahdat et al. (2021) combine the advantages of DDPMs and Variational Autoencoders and get better results. Furthermore, Kim et al. (2021) try to solve an existing bottleneck that the inference equations of DDPMs are unbounded in some situations.

Song et al. (2020b) find the similarity between DDPMs and noise conditional score networks (NCSNs (Song & Ermon, 2020b)), which is that they both use a process similar to Langevin dynamics to produce samples. Therefore, some works (Song & Ermon, 2020a; Kim et al., 2021) that can improve the results of NCSNs also can be used in DDPMs. Additional, Song et al. (2020b) combine DDPMs and NCSNs under the framework of neural differential equations (Chen et al., 2019; Dupont et al., 2019). Therefore, numerical methods widely used in neural differential equations can also be applied to accelerate DDPMs. Our work successfully combines the advantages of Song et al. (2020a) and Song et al. (2020b). We use a transfer part from DDIMs and use different gradient parts from different numerical methods. Although we and Song et al. (2020b) solve certain differential

equations derived from DDPMs, we use different target different equations and different numerical methods, which get better results.

The application of DDPMs is not limited to unconditional image generation. Some works apply DDPMs to various types of data successfully, including text-to-speech (Chen et al., 2020; Lam et al., 2021), singing voice (Liu et al., 2021), 3D Point Cloud (Luo & Hu, 2021), text generation (Austin et al., 2021). Additionally, DDPMs can also be used to generate conditional samples, too (Jeong et al., 2021; Choi et al., 2021).

## A.2 CONVERGENT ORDER OF METHOD

We use the forward Euler method and linear multi-step method to show what is the order of a method. Assume that $x_t$ is precise and compute the error at $x_{t+\delta}$. For forward Euler method, we have:

$$
\begin{aligned}
e_{t,\delta} &= x(t+\delta) - x_{t,\delta} \\
&= \left( x(t) + \delta f(x(t), t) + \frac{\delta^2}{2} f''(c) \right) - (x(t) + \delta f(x_t, t)) \\
&= \frac{\delta^2}{2} f''(c) \leq \frac{\delta^2}{2} M
\end{aligned}
\tag{15}
$$

Here, we assume that $f''$ is continuous, so $f''$ is bounded in a close area. $x(t+\delta)$ is the precise result and $x_{t,\delta}$ is the numerical result from $t$ to $t+\delta$.

For linear multi-step method, we have:

$$
\begin{aligned}
e_{t,\delta} =&\, x(t+\delta) - x_{t,\delta} \\
=& \left( x(t) + \frac{\delta}{1!} f(x(t), t) + \frac{\delta^2}{2!} f'(x(t), t) + \cdots + \frac{\delta^4}{4!} f^{(3)}(x(t), t) + O\left(\delta^5\right) \right) \\
& - \left( x(t) + b_1 \delta f(x(t), t) + \sum_{s=2}^{4} b_s \delta \left( \sum_{k=0}^{4} \frac{(-(s-1)\delta)^k}{(k)!} f^{(k-1)}(x(t), t) + O\left(\delta^5\right) \right) \right) \\
=&\, O(\delta^5)
\end{aligned}
\tag{16}
$$

Here, $\{b_s\}$ satisfy $\sum_{s=1}^{4} b_s = 1$ and the following equations for $j \in \{1, \cdots, 3\}$:

$$
(-1)^j b_2 + (-2)^j b_3 + (-3)^j b_4 = \frac{1}{j+1}.
\tag{17}
$$

We call the error at $x_{t,\delta}$ local error, and the error at $x_{t+M\delta}$ (M is big enough but finite) global error. Assume the local error of our method has order k+1 and the target ODE satisfies Lipschitz condition, then:

$$
\begin{aligned}
x(t+M\delta) - x_{t,M\delta} \leq&\, |x(t+M\delta) - x_{t+(M-1)\delta,\delta}| + |x_{t+(M-1)\delta,\delta} - x_{t,M\delta}| \\
\leq&\, e_{t+(M-1)\delta,\delta} + e^{L\delta} |x(t+(M-1)\delta) - x_{t,(M-1)\delta}| \\
\leq&\, e_{t+(M-1)\delta,\delta} + e^{L\delta} e_{t+(M-2)\delta,\delta} + \cdots \\
\leq&\, C\delta^{k+1} \left( 1 + e^{Lh} + \cdots + e^{(i-1)Lh} \right) \\
=&\, Ch^{k+1} \frac{e^{iL\delta} - 1}{e^{L\delta} - 1} \leq C\delta^{k+1} \frac{e^{iL\delta} - 1}{L\delta} \\
=&\, O(\delta^k).
\end{aligned}
\tag{18}
$$

Therefore, the global error will be one order lower than the local error. From Equation (15), we can see the forward Euler method has local error $O(\delta^2)$ and global error $O(\delta)$, so we call it the first-order numerical method. And the linear multi-step method has local error $O(\delta^5)$ and global error $O(\delta^4)$, and we call it the fourth-order numerical method.

In addition, assuming that a numerical method has kth-order global error, we can compute the convergence speed of this numerical method:

$$
\lim_{\delta \to 0} \frac{x_{t+T}^{2\delta} - x(t+T)}{x_{t+T}^{\delta} - x(t+T)} = \frac{(2\delta)^k}{\delta^k} = 2^k.
\tag{19}
$$

Here, $x_{t+T}^{\delta}$ is the result at t+T and move $\delta$ every step. This shows that the fourth-order method can converge to the exact solution faster than the first-order method when $\delta \to 0$, which means that we can use a bigger iteration interval $\delta$ to achieve similar global error and a bigger iteration interval means that we can iterate fewer times to get results with high quality.

### A.3 PSEUDO SECOND-ORDER METHOD

We introduce two second-order numerical methods. First is improved Euler method satisfying:

$$\begin{cases} k_1 = f(x_t, t) \\ k_2 = f(x_t + \delta k_1, t + \delta) \\ x_{t+\delta} = x_t + \frac{\delta}{2}(k_1 + k_2) \end{cases} \tag{20}$$

Second is another linear multi-step method called second-order linear multi-step method satisfying:

$$x_{t+\delta} = x_t + \frac{\delta}{2}(3f_t - f_{t-\delta}) \tag{21}$$

And the corresponding pseudo improved Euler methods satisfying:

$$\begin{cases} e_t^1 = \epsilon_\theta(x_t, t) \\ x_t^1 = \phi(x_t, e_t^1, t, t+\delta) \\ e_t^2 = \epsilon_\theta(x_t^1, t+\delta) \\ e_t' = \frac{1}{2}(e_t^1 + e_t^2) \\ x_{t+\delta} = \phi(x_t, e_t', t, t+\delta) \end{cases} \tag{22}$$

Pseudo second-order linear multi-step method satisfying:

$$\begin{cases} e_t = \epsilon_\theta(x_t, t) \\ e_t' = \frac{1}{2}(3e_t - e_{t-\delta}) \\ x_{t+\delta} = \phi(x_t, e_t', t, t+\delta) \end{cases} \tag{23}$$

Similar to what we do to get F-PNDMs, We combine them to get S-PNDMs. Abbreviate Equation (22) and (23) as

$$x_{t+\delta}, e_t = PIE(x_t, \{e_p\}_{p<t}, t, t+\delta),$$
$$x_{t+\delta}, e_t^1 = PLMS'(x_t, t, t+\delta).$$

**Algorithm 3** S-PNDMs

1: $x_T \sim \mathcal{N}(0, I)$
2: **for** $t = T - 1$ **do**
3:    $x_t, e_t = PIE(x_{t+1}, t+1, t)$
4: **end for**
5: **for** $t = T - 2, \cdots, 1, 0$ **do**
6:    $x_t, e_t = PLMS'(x_{t+1}, \{e_p\}_{p>t}, t+1, t)$
7: **end for**
8: **return** $x_0$

### A.4 THE EXISTENCE OF A DERIVATIVE

Because $\bar{\alpha}_t$ is usually obtained by multiplying a linear variance schedule $\beta_t$. So we have

$$\bar{\alpha}_t = e^{at^2 + bt + c}, \tag{24}$$

and $\bar{\alpha}_0 = 1$, so $c = 0$. Now we have

$$\begin{aligned}
&\lim_{\delta \to 0} \frac{x_{t-\delta} - x_t}{\delta} \\
&= \lim_{\delta \to 0} \frac{\bar{\alpha}_{t-\delta} - \bar{\alpha}_t}{\delta} \left( \frac{x_t}{\sqrt{\bar{\alpha}_t}(\sqrt{\bar{\alpha}_{t-\delta}} + \sqrt{\bar{\alpha}_t})} - \frac{\epsilon_\theta(x_t, t)}{\sqrt{\bar{\alpha}_t}(\sqrt{(1-\bar{\alpha}_{t-\delta})\bar{\alpha}_t} + \sqrt{(1-\bar{\alpha}_t)\bar{\alpha}_{t-\delta}})} \right) \\
&= \lim_{\delta \to 0} \frac{\bar{\alpha}_{t-\delta} - \bar{\alpha}_t}{\delta} \left( \frac{x_t}{2\bar{\alpha}_t} - \frac{\epsilon_\theta(x_t, t)}{2\sqrt{1-\bar{\alpha}_t}\bar{\alpha}_t} \right) = (e^{at^2+bt})' \left( \frac{x_t}{2\bar{\alpha}_t} - \frac{\epsilon_\theta(x_t, t)}{2\sqrt{1-\bar{\alpha}_t}\bar{\alpha}_t} \right) \\
&= (2at + b)\bar{\alpha}_t \left( \frac{x_t}{2\bar{\alpha}_t} - \frac{\epsilon_\theta(x_t, t)}{2\sqrt{1-\bar{\alpha}_t}\bar{\alpha}_t} \right) = \frac{1}{2}(2at+b)(x_t - \frac{\epsilon_\theta(x_t, t)}{\sqrt{1-\bar{\alpha}_t}}).
\end{aligned} \tag{25}$$

To make $\lim_{\delta \to 0} \frac{x_{t-\delta} - x_t}{\delta}|_{t=0}$ is well-defined, $b$ must equal to zero, or $(2at + b)\frac{\epsilon_\theta(x_t,t)}{\sqrt{1-\bar{\alpha}_t}}$ will tend to infinity. This is a strong condition that most variance schedules do not satisfy. In practice, DDPMs can choose the variance schedule very freely. This means that treating DDPMs as ODEs directly is not proper and has theoretical weakness.

## A.5 RELATIONSHIP BETWEEN $t$, $\epsilon_\theta$ AND $x_t$

**Relationship between $t$ and $\epsilon_\theta$.** In Figure 5, we can see that the denoising process tends to converge, whether in the $\epsilon_\theta$ domain or the sample/image domain when the step-index tends to zero. Therefore, we can say that the noise becomes more and more precise when step, namely $t$, tends to zero.

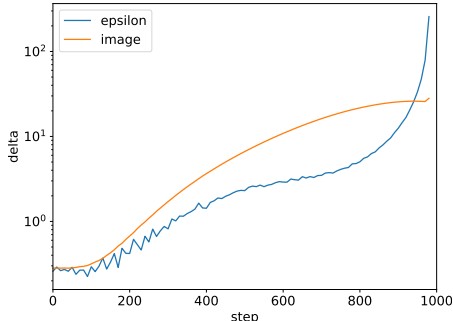

Figure 5: The norm $\delta$ of the difference between two adjacent terms under different steps

**Relationship between $\epsilon_\theta$ and $x_t$** To prove Property 3.1, assume that $x_t = \sqrt{\bar{\alpha}_t}x_0 + \sqrt{1-\bar{\alpha}_t}\epsilon$, NN is the neural network and $\epsilon_\theta = \text{NN}(x_t, t)$. Because we assume that the gradient part is precise, then we have $\epsilon_\theta = \epsilon$. Then for all $t' \leq t$, we have:

$$
\begin{aligned}
x_{t'} &= \sqrt{\bar{\alpha}_{t'}} \left( \frac{x_t - \sqrt{1-\bar{\alpha}_t}\epsilon_\theta}{\sqrt{\bar{\alpha}_t}} \right) + \sqrt{1-\bar{\alpha}_{t'}}\epsilon_\theta \\
&= \sqrt{\bar{\alpha}_{t'}} \left( \frac{\sqrt{\bar{\alpha}_t}x_0 + \sqrt{1-\bar{\alpha}_t}\epsilon - \sqrt{1-\bar{\alpha}_t}\epsilon}{\sqrt{\bar{\alpha}_t}} \right) + \sqrt{1-\bar{\alpha}_{t'}}\epsilon_\theta(x_t, t) \\
&= \sqrt{\bar{\alpha}_{t'}}x_0 + \sqrt{1-\bar{\alpha}_{t'}}\epsilon.
\end{aligned}
\tag{26}
$$

Here, we can find that $x_{t'} = \sqrt{\bar{\alpha}_{t'}}x_0 + \sqrt{1-\bar{\alpha}_{t'}}\epsilon$ is also precise, so Property 3.1 is true.

## A.6 ORDER ANALYSIS OF PSEUDO METHOD

For the convenience of theoretical analysis, we generalize the problem. Let $\phi(x(t), \epsilon, t, \delta) = f(x(t), t, \delta) + g(t, \delta)\epsilon(x(t), t)$ and we have the property $f(x(t), t, 0) = g(t, 0) = 0$. Then we have:

$$
\begin{aligned}
x(1) &= x(0) + \sum_{\delta \to 0} (y(t+\delta) - y(t)) \\
&= x(0) + \sum_{\delta \to 0} (f(x(t), t, \delta) + g(t, \delta)\epsilon(x(t), t)) \\
&= x(0) + \int_0^1 \left( \frac{\partial f}{\partial \delta}(x(t), t, 0) + \frac{\partial g}{\partial \delta}(t, 0)\epsilon(x(t), t) \right).
\end{aligned}
\tag{27}
$$

Now, Equation (10) becomes a special case of this more general version and, in this special case, we have:

$$
\begin{aligned}
f(x(t), t, \delta) &= \left( \frac{\sqrt{\alpha(t+\delta)}}{\sqrt{\alpha}} - 1 \right) x(t) \\
g(t, \delta) &= \sqrt{1 - \alpha(t+\delta)} - \frac{\sqrt{(1-\alpha(t))\alpha(t+\delta)}}{\sqrt{\alpha(t)}}
\end{aligned}
\tag{28}
$$

Now, we compute the local error of S-PNDMs. We first compute the theoretical and numerical results of different numerical methods. We have:

$$
\begin{aligned}
&x(t+\delta) \\
=&x(t) + \delta\left(\frac{\partial f}{\partial \delta}(x(t),t,0) + \frac{\partial g}{\partial \delta}(t,0)\epsilon(x(t),t)\right) + \\
&\frac{\delta^2}{2}\left(\frac{\partial f}{\partial \delta}(x(t),t,0) + \frac{\partial g}{\partial \delta}(t,0)\epsilon(x(t),t)\right)' + O(\delta^3) \\
=&x(t) + \delta\left(\frac{\partial f}{\partial \delta}(x(t),t,0) + \frac{\partial g}{\partial \delta}(t,0)\epsilon(x(t),t)\right) + O(\delta^3) + \\
&\frac{\delta^2}{2}\left(\frac{\partial^2 f}{\partial \delta \partial t}(x(t),t,0) + \frac{\partial^2 f}{\partial \delta \partial x}(x(t),t,0)\left(\frac{\partial f}{\partial \delta}(x(t),t,0) + \frac{\partial g}{\partial \delta}(t,0)\epsilon(x(t),t)\right)\right) + \\
&\frac{\delta^2}{2}\left(\frac{\partial^2 g}{\partial \delta \partial t}(t,0)\epsilon(x(t),t) + \frac{\partial g}{\partial \delta}(t,0)\epsilon'(x(t),t)\right)
\end{aligned}
\tag{29}
$$

and

$$
\begin{aligned}
&x_{\text{S-PNDM}}(t+\delta) \\
=&x(t) + f(x(t),t,\delta) + g(t,\delta)\frac{1}{2}(\epsilon(x(t),t) + \epsilon(x(t) + \phi(...,t,\delta),t+\delta)) \\
=&x(t) + \delta\frac{\partial f}{\partial \delta}(x(t),t,0) + \frac{\delta^2}{2}\frac{\partial^2 f}{\partial \delta^2}(x(t),t,0) + O(\delta^3) + \\
&\left(\delta\frac{\partial g}{\partial \delta}(t,0) + \frac{\delta^2}{2}\frac{\partial^2 g}{\partial \delta^2}(t,0)\right)\frac{1}{2}(\epsilon(x(t),t) + \epsilon(x(t+\delta) + O(\delta^2),t+\delta)) \\
=&x(t) + \delta\frac{\partial f}{\partial \delta}(x(t),t,0) + \frac{\delta^2}{2}\frac{\partial^2 f}{\partial \delta^2}(x(t),t,0) + O(\delta^3) + \\
&\left(\delta\frac{\partial g}{\partial \delta}(t,0) + \frac{\delta^2}{2}\frac{\partial^2 g}{\partial \delta^2}(t,0)\right)\frac{1}{2}(\epsilon(x(t),t) + \epsilon(x(t+\delta),t+\delta)) \\
=&x(t) + \delta\frac{\partial f}{\partial \delta}(x(t),t,0) + \frac{\delta^2}{2}\frac{\partial^2 f}{\partial \delta^2}(x(t),t,0) + O(\delta^3) + \\
&\left(\delta\frac{\partial g}{\partial \delta}(t,0) + \frac{\delta^2}{2}\frac{\partial^2 g}{\partial \delta^2}(t,0)\right)\frac{1}{2}(\epsilon(x(t),t) + \epsilon(x(t),t) + \delta\epsilon(x(t),t)') \\
=&x(t) + \delta\frac{\partial f}{\partial \delta}(x(t),t,0) + \frac{\delta^2}{2}\frac{\partial^2 f}{\partial \delta^2}(x(t),t,0) + O(\delta^3) + \\
&\delta\frac{\partial g}{\partial \delta}(t,0)\left(\epsilon(x(t),t) + \frac{1}{2}\delta\epsilon(x(t),t)'\right) + \frac{\delta^2}{2}\frac{\partial^2 g}{\partial \delta^2}(t,0)\epsilon(x(t),t) \\
=&x(t) + \delta\left(\frac{\partial f}{\partial \delta}(x(t),t,0) + \frac{\partial g}{\partial \delta}(t,0)\epsilon(x(t),t)\right) + O(\delta^3) + \\
&\frac{\delta^2}{2}\left(\frac{\partial^2 f}{\partial \delta^2}(x(t),t,0) + \frac{\partial g}{\partial \delta}(t,0)\epsilon(x(t),t)' + \frac{\partial^2 g}{\partial \delta^2}(t,0)\epsilon(x(t),t)\right).
\end{aligned}
\tag{30}
$$

Then we compute the difference between the theoretical and numerical results. We have:

$$
\begin{aligned}
&x(t+\delta) - x_{\text{S-PNDM}}(x+\delta) \\
=&\frac{\delta^2}{2}\left((\frac{\partial^2 f}{\partial \delta \partial t} - \frac{\partial^2 f}{\partial \delta^2})(x(t),t,0) + \frac{\partial^2 f}{\partial \delta \partial x}(x(t),t,0)\frac{\partial f}{\partial \delta}(x(t),t,0)\right) + \\
&\frac{\delta^2}{2}\left((\frac{\partial^2 g}{\partial \delta \partial t} - \frac{\partial^2 g}{\partial \delta^2})(t,0) + \frac{\partial^2 f}{\partial \delta \partial x}(x(t),t,0)\frac{\partial g}{\partial \delta}(t,0)\right)\epsilon(x(t),t) + O(\delta^3)
\end{aligned}
\tag{31}
$$

In this special case, we compute the derivatives of some items needed in Equation (31). We have:

$$\frac{\partial g}{\partial \delta}(t,\delta) = \frac{\partial}{\partial \delta}\left(\sqrt{1-\alpha(t+\delta)} - \frac{\sqrt{(1-\alpha(t))\alpha(t+\delta)}}{\sqrt{\alpha(t)}}\right)$$

$$= \frac{-\alpha'(t+\delta)}{2\sqrt{1-\alpha((t+\delta))}} - \frac{\sqrt{1-\alpha(t)}\alpha'(t+\delta)}{2\sqrt{\alpha(t)\alpha(t+\delta)}} \tag{32}$$

$$\frac{\partial f}{\partial \delta}(x(t),t,\delta) = \frac{\partial}{\partial \delta}\left(\left(\frac{\sqrt{\alpha(t+\delta)}}{\sqrt{\alpha}}-1\right)x(t)\right)$$

$$= \frac{\alpha'(t+\delta)}{2\sqrt{\alpha(t)\alpha(t+\delta)}}x(t) \tag{33}$$

$$\frac{\partial^2 f}{\partial \delta^2}(x(t),t,\delta)|_{\delta=0} = \frac{\alpha''(t+\delta)}{2\sqrt{\alpha(t)\alpha(t+\delta)}}x(t) + \frac{-\alpha'(t+\delta)^2}{4\sqrt{\alpha(t)\alpha(t+\delta)^3}}x(t)|_{\delta=0} \tag{34}$$

$$\frac{\partial^2 f}{\partial \delta \partial t}(x(t),t,0) = \frac{\alpha''(t)}{2\alpha(t)}x(t) - \frac{\alpha'(t)^2}{2\alpha(t)^2}x(t) \tag{35}$$

$$\frac{\partial^2 f}{\partial \delta \partial x}(x(t),t,0) = \frac{\alpha'(t)}{2\alpha(t)} \tag{36}$$

$$\frac{\partial^2 g}{\partial \delta^2}(t,\delta)|_{\delta=0} = \frac{-\alpha''(t+\delta)}{2\sqrt{1-\alpha((t+\delta))}} - \frac{\sqrt{1-\alpha(t)}\alpha''(t+\delta)}{2\sqrt{\alpha(t)\alpha(t+\delta)}} +$$

$$\frac{-\alpha'(t+\delta)^2}{4\sqrt{1-\alpha(t+\delta)}^3} + \frac{\sqrt{1-\alpha(t)}\alpha'(t+\delta)^2}{4\sqrt{\alpha(t)\alpha(t+\delta)^3}}|_{\delta=0}$$

$$= \frac{-\alpha''(t)}{2\sqrt{1-\alpha((t))}} - \frac{\sqrt{1-\alpha(t)}\alpha''(t)}{2\sqrt{\alpha(t)\alpha(t)}} + \tag{37}$$

$$\frac{-\alpha'(t)^2}{4\sqrt{1-\alpha(t)}^3} + \frac{\sqrt{1-\alpha(t)}\alpha'(t)^2}{4\sqrt{\alpha(t)\alpha(t)^3}}$$

$$\frac{\partial^2 g}{\partial \delta \partial t}(t,\delta)|_{\delta=0} = \frac{\partial}{\partial t}\left(\frac{-\alpha'(t)}{2\sqrt{1-\alpha((t))}} - \frac{\sqrt{1-\alpha(t)}\alpha'(t)}{2\sqrt{\alpha(t)\alpha(t)}}\right)$$

$$= \frac{-\alpha''(t)}{2\sqrt{1-\alpha((t))}} - \frac{\sqrt{1-\alpha(t)}\alpha''(t)}{2\sqrt{\alpha(t)\alpha(t)}} + \tag{38}$$

$$\frac{-\alpha'(t)^2}{4\sqrt{1-\alpha(t)}^3} + \frac{\sqrt{1-\alpha(t)}\alpha'(t)^2}{2\alpha(t)^2} + \frac{\alpha'(t)^2}{4\sqrt{\alpha^2(t)(1-\alpha(t))}}$$

Now we can compute the final result of Equation (31). We split it into three parts and the values of the first two terms. We have:

$$(\frac{\partial^2 f}{\partial \delta \partial t} - \frac{\partial^2 f}{\partial \delta^2})(x(t),t,0) + \frac{\partial^2 f}{\partial \delta \partial x}(x(t),t,0)\frac{\partial f}{\partial \delta}(x(t),t,0)$$

$$= \left(\frac{\alpha''(t)}{2\alpha(t)}x(t) - \frac{\alpha'(t)^2}{2\alpha(t)^2}x(t)\right) - \left(\frac{\alpha''(t)}{2\alpha(t)}x(t) + \frac{-\alpha'(t)^2}{4\alpha(t)^2}x(t)\right) + \frac{1}{2\alpha(t)}\frac{\alpha'(t)}{2\alpha(t)}x(t) \tag{39}$$

$$=0$$

and

$$(\frac{\partial^2 g}{\partial \delta \partial t} - \frac{\partial^2 g}{\partial \delta^2})(t,0) + \frac{\partial^2 f}{\partial \delta \partial x}(x(t),t,0)\frac{\partial g}{\partial \delta}(t,0)$$

$$= \frac{\sqrt{1-\alpha(t)}\alpha'(t)^2}{4\alpha(t)^2} + \frac{\alpha'(t)^2}{4\alpha(t)\sqrt{1-\alpha(t)}} + \frac{\alpha'(t)}{2\alpha(t)}\left(\frac{-\alpha'(t)}{2\sqrt{1-\alpha((t))}} - \frac{\sqrt{1-\alpha(t)}\alpha'(t)}{2\alpha(t)}\right) \tag{40}$$

$$= \frac{\alpha'(t)^2}{4\alpha(t)^2\sqrt{1-\alpha(t)}} + \frac{\alpha'(t)}{2\alpha(t)}\left(\frac{-\alpha'(t)}{2\sqrt{1-\alpha(t)}\alpha(t)}\right)$$

$$=0$$

Finally, we get the final result of Equation (31):

$$x(t + \delta) - x_{\text{S-PNDM}}(x + \delta) = O(\delta^3) \tag{41}$$

And the computation of the convergence order of F-PNDMs is similar, and we ignore it here. Therefore, Property 3.2 is true.

### A.7 VARIANCE SCHEDULE

According to Cifar10 (cosine) in Table 2, PNDMs can be used on both linear variance schedule and cosine variance schedule. However, we also notice cosine variance schedule can make FID lower when we use relatively big generation steps, but the effort is limited when the number of steps is small. F-PNDM uses information from four consecutive steps, so the smoothness of the schedule is more important for F-PNDM than DDIM. According to this experiment, our work can be used with works that pay attention to variance schedules to improve the acceleration effect further. However, a variance schedule that fits pseudo numerical methods better remains to be found in further work.

### A.8 TOY EXAMPLE

Here, we design a toy example to test our new methods without the influence of neural networks. We randomly generate the initial input $x_1 = (m_1, m_2)$, $m_i \sim U(0, 1)$ and use a simple analytic equations $\epsilon_\theta(x) = (\sin x[0], \cos x[1])$ to replace the neural networks in real cases. Let $\phi$ in Equation (11) is unchanged and $\bar{\alpha}_t = \alpha(t) = 1 - t$, then we get:

$$
\begin{aligned}
&\phi(x_t, \epsilon_\theta(x_t), t, t - \delta) \\
=&\frac{\sqrt{\bar{\alpha}_{t-\delta}}}{\sqrt{\bar{\alpha}_t}} x_t - \frac{(\bar{\alpha}_{t-\delta} - \bar{\alpha}_t)}{\sqrt{\bar{\alpha}_t}(\sqrt{(1 - \bar{\alpha}_{t-\delta})\bar{\alpha}_t} + \sqrt{(1 - \bar{\alpha}_t)\bar{\alpha}_{t-\delta}})} \epsilon_t \\
=&\frac{\sqrt{1 - (t - \delta)}}{\sqrt{1 - t}} x_t - \frac{\delta}{\sqrt{1 - t}\left(\sqrt{(t - \delta)(1 - t)} + \sqrt{t(1 - (t - \delta))}\right)} \epsilon_\theta(x_t)
\end{aligned}
\tag{42}
$$

Here, we use three different numerical methods to generate $x_0$.

For DDIM, we have:

$$x_{t-\delta} = x_t + \phi(x_t, \epsilon_\theta(x_t), t, t - \delta) \tag{43}$$

For FON, according to Equation (10), we have:

$$
\begin{aligned}
e_t' &= \bar{\alpha}'(t)\left(\frac{x_t}{2\bar{\alpha}(t)} - \frac{\epsilon_\theta(x_t)}{2\bar{\alpha}(t)\sqrt{1 - \bar{\alpha}(t)}}\right) \\
&= -\left(\frac{x_t}{2(1 - t)} - \frac{\epsilon_\theta(x_t)}{2(1 - t)\sqrt{t}}\right) \\
x_{t-\delta} &= x_t + \frac{\delta}{24}(55e_t' - 59e_{t+\delta}' + 37e_{t+2\delta}' - 9e_{t+3\delta}')
\end{aligned}
\tag{44}
$$

For F-PNDM, we have:

$$
\begin{aligned}
e' &= \frac{1}{24}(55\epsilon_\theta(x_t) - 59\epsilon_\theta(x_{t+\delta}) + 37\epsilon_\theta(x_{t+2\delta}) - 9\epsilon_\theta(x_{t+3\delta})) \\
x_{t-\delta} &= x_t + \phi(x_t, e', t, t - \delta)
\end{aligned}
\tag{45}
$$

Then we draw the corresponding generation curves in Figure 6. We find that the result is similar to the real cases. The main difference here is that FON can correct its results while the real case cannot. The reason is that the gradient is well-defined everywhere, while in real cases, the gradient is meaningful on the high-density region of the data $x_t$ of DDPMs.

### A.9 MORE FID RESULTS

Here, We provide our more detailed FID results on Cifar10, CelebA, LSUN-church and LSUN-bedroom in Table 3, 4, 5, 6.

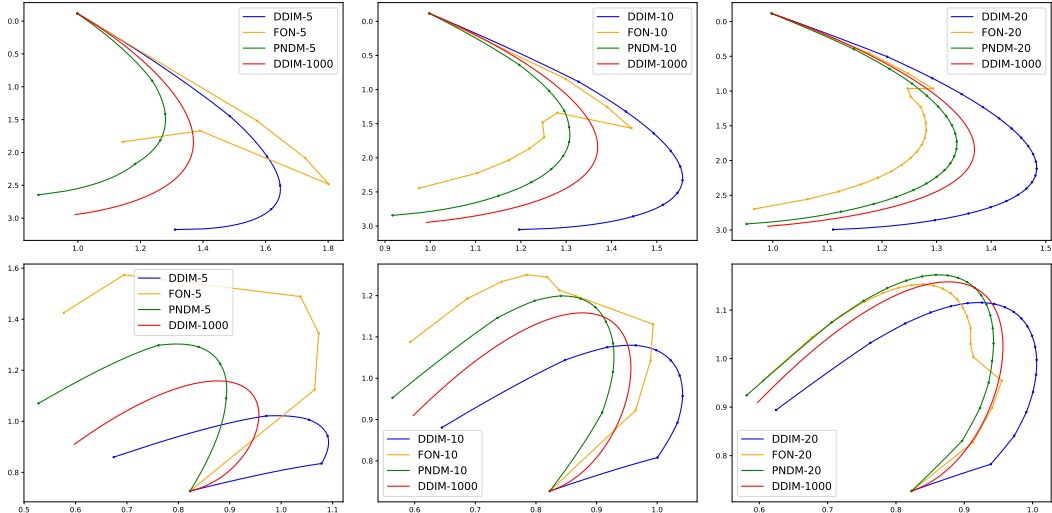

Figure 6: The generation curve of our toy example.

| steps | 5 | 10 | 20 | 25 | 40 | 50 | 100 | 125 | 200 | 250 | 500 | 1000 |
|---|---|---|---|---|---|---|---|---|---|---|---|---|
| DDIM | | 13.4 | | 6.84 | | 4.67 | 4.16 | | | | | 4.04 |
| DDIM* | 44.5 | 18.5 | 10.9 | 9.61 | 7.65 | 6.99 | 5.52 | 5.19 | 4.69 | 4.52 | 4.17 | 4.00 |
| FON | 98.0 | 13.1 | 7.41 | 6.41 | 5.50 | 5.26 | 4.65 | 4.54 | 4.23 | 4.12 | 3.84 | 3.71 |
| S-PNDM | 22.8 | 11.6 | 7.56 | 6.79 | 5.57 | 5.18 | 4.34 | 4.18 | 3.97 | 3.91 | 3.81 | 3.80 |
| F-PNDM | 13.9 | 7.03 | 5.00 | 4.76 | 4.10 | 3.95 | 3.72 | 3.64 | 3.60 | **3.60** | 3.64 | 3.70 |
| DDIM* | 28.7 | 14.5 | 8.79 | 7.83 | 6.41 | 5.86 | 4.92 | 4.75 | 4.42 | 4.30 | 3.98 | 3.69 |
| S-PNDM | 18.3 | 8.64 | 5.77 | 5.45 | 4.76 | 4.46 | 3.94 | 3.85 | 3.69 | 3.71 | 3.60 | 3.38 |
| F-PNDM | 18.2 | 7.05 | 4.61 | 4.32 | 3.85 | 3.68 | 3.53 | 3.46 | 3.47 | 3.49 | 3.44 | **3.26** |

Table 3: Cifar10 image generation measured in FID. The upper part uses linear variance schedule and the bottom half uses cosine variance schedule. The first line shows the FID provided by Song et al. (2020a).

| steps | 5 | 10 | 20 | 25 | 40 | 50 | 100 | 125 | 200 | 250 | 500 | 1000 |
|---|---|---|---|---|---|---|---|---|---|---|---|---|
| DDIM | | 17.3 | | 13.7 | | 9.17 | 6.53 | | | | | 3.51 |
| DDIM* | 24.4 | 16.9 | 13.4 | 12.3 | 9.99 | 8.95 | 6.36 | 5.74 | 4.78 | 4.44 | 3.75 | 3.41 |
| FON | 60.2 | 16.0 | 11.6 | 10.6 | 8.89 | 8.13 | 6.70 | 6.28 | 5.45 | 5.14 | 4.49 | 4.17 |
| S-PNDM | 15.2 | 12.2 | 9.45 | 8.42 | 6.50 | 5.69 | 4.03 | 3.72 | 3.30 | 3.19 | 3.01 | 2.99 |
| F-PNDM | 11.3 | 7.71 | 5.51 | 4.75 | 3.67 | 3.34 | 2.81 | 2.75 | **2.71** | 2.71 | 2.77 | 2.86 |

Table 4: CelebA image generation measured in FID. All of them use linear variance schedule.

| steps | 5 | 10 | 20 | 25 | 40 | 50 | 100 | 125 | 200 | 250 |
|---|---|---|---|---|---|---|---|---|---|---|
| DDIM | | 19.5 | 12.5 | | | 10.8 | 10.6 | | | |
| DDIM* | 48.8 | 18.8 | 11.7 | 11.0 | 10.1 | 10.0 | 9.84 | 9.83 | 9.85 | 9.88 |
| S-PNDM | 20.5 | 11.8 | 9.20 | 9.13 | 9.31 | 9.49 | 9.82 | 9.88 | 10.0 | 10.0 |
| F-PNDM | 14.8 | **8.69** | 9.13 | 9.33 | 9.69 | 9.89 | 10.1 | 9.99 | 10.1 | 10.1 |

Table 5: LSUN-church image generation measured in FID. All of them use linear variance schedule.

| steps | 5 | 10 | 20 | 25 | 40 | 50 | 100 | 125 | 200 | 250 |
|---|---|---|---|---|---|---|---|---|---|---|
| DDIM | | 17.0 | 8.89 | | | 6.75 | 6.62 | | | |
| DDIM* | 51.3 | 16.4 | 8.47 | 7.41 | 6.27 | 6.05 | 5.97 | 6.03 | 6.23 | 6.32 |
| S-PNDM | 18.1 | 10.2 | 6.50 | 6.02 | 5.74 | 5.81 | 6.29 | 6.44 | 6.69 | 6.75 |
| F-PNDM | 12.6 | 6.99 | **5.68** | 5.74 | 6.17 | 6.44 | 6.91 | 6.96 | 7.03 | 6.92 |

Table 6: LSUN-bedroom image generation measured in FID. All of them use linear variance schedule.

### A.10 MORE IMAGE RESULTS

Here, we show more generated images on Cifar10, CelebA, LSUN-church and LSUN-bedroom in Figure 7, 9, 8, 10, 11, 12.

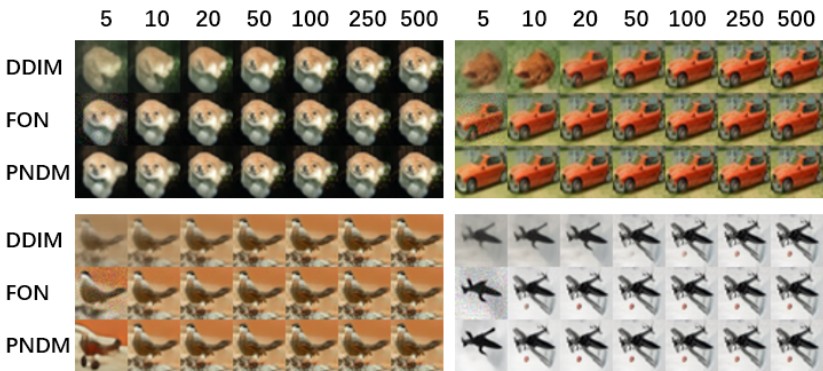

Figure 7: 5, 10, 20, 50, 100, 250, 500-steps generated results using DDIMs, classical numerical methods and PNDMs on Cifar10.

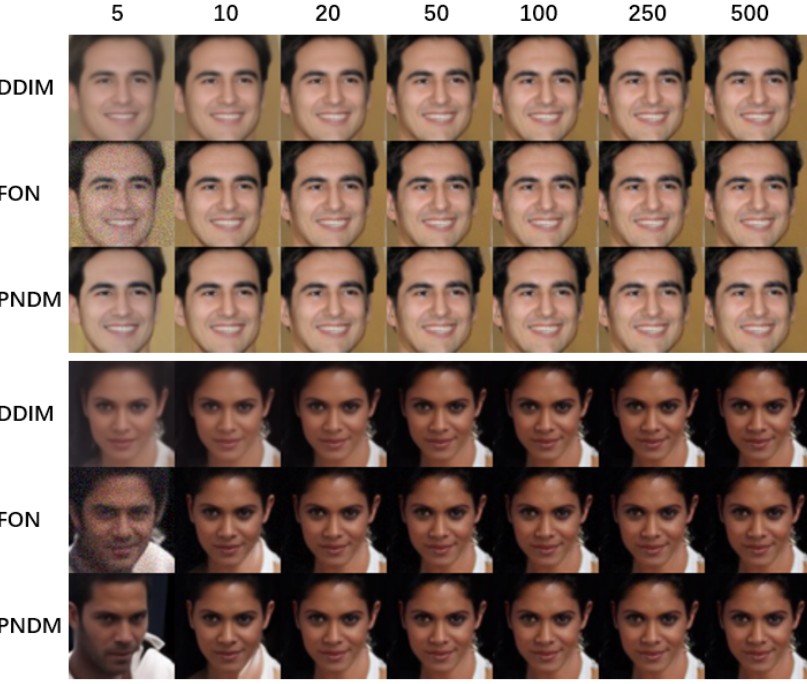

Figure 8: 5, 10, 20, 50, 100, 250, 500-steps generated results using DDIMs, classical numerical methods and PNDMs on CelebA.

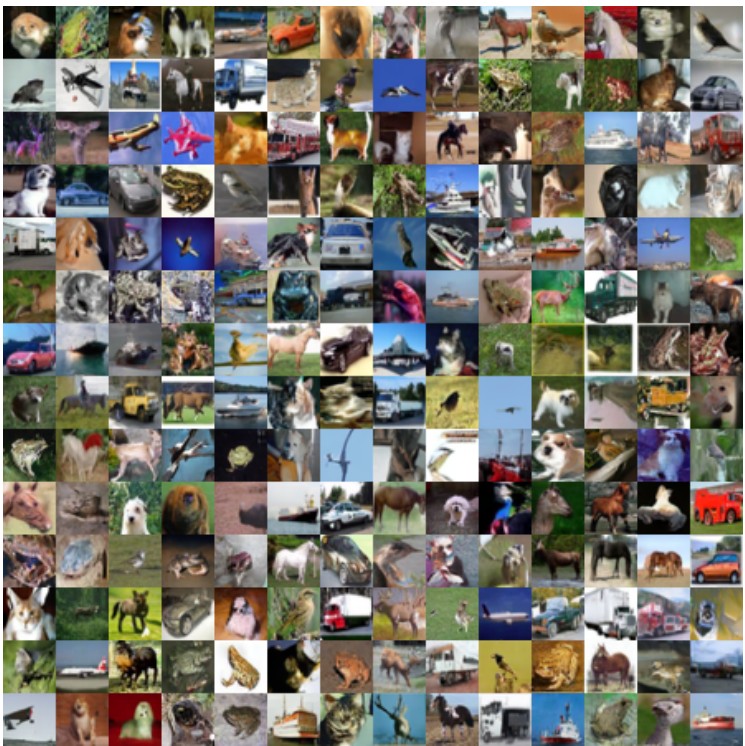

Figure 9: Generated images of PNDMs on Cifar10.

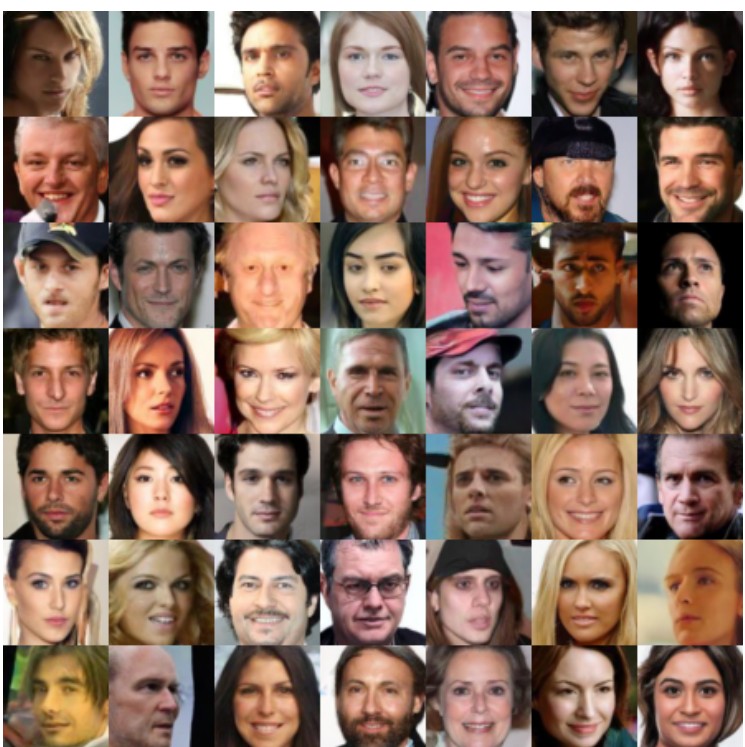

Figure 10: Generated images of PNDMs on CelebA.

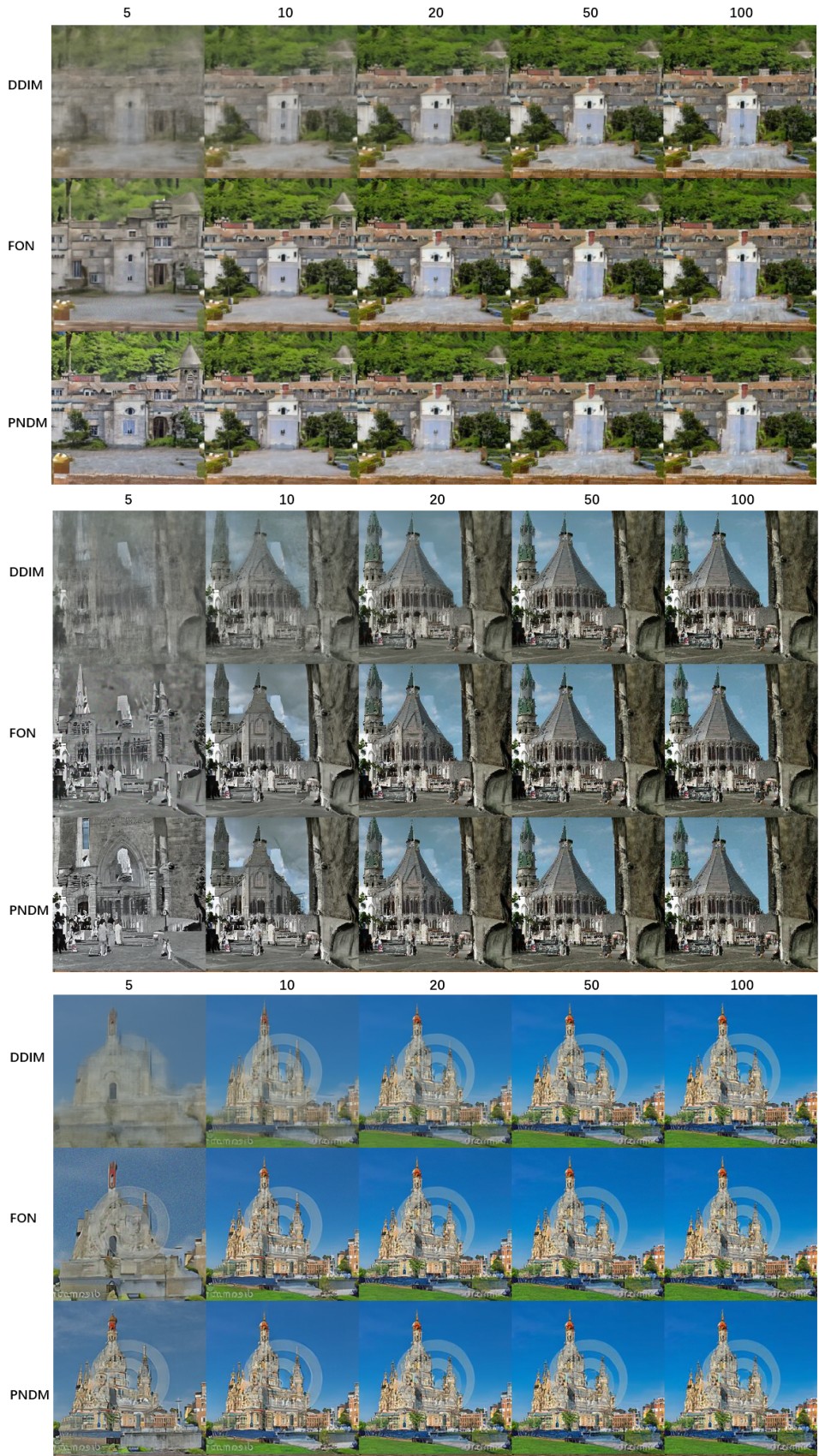

Figure 11: 5, 10, 20, 50, 100-steps generated results using DDIMs, classical numerical methods and PNDMs on LSUN-church.

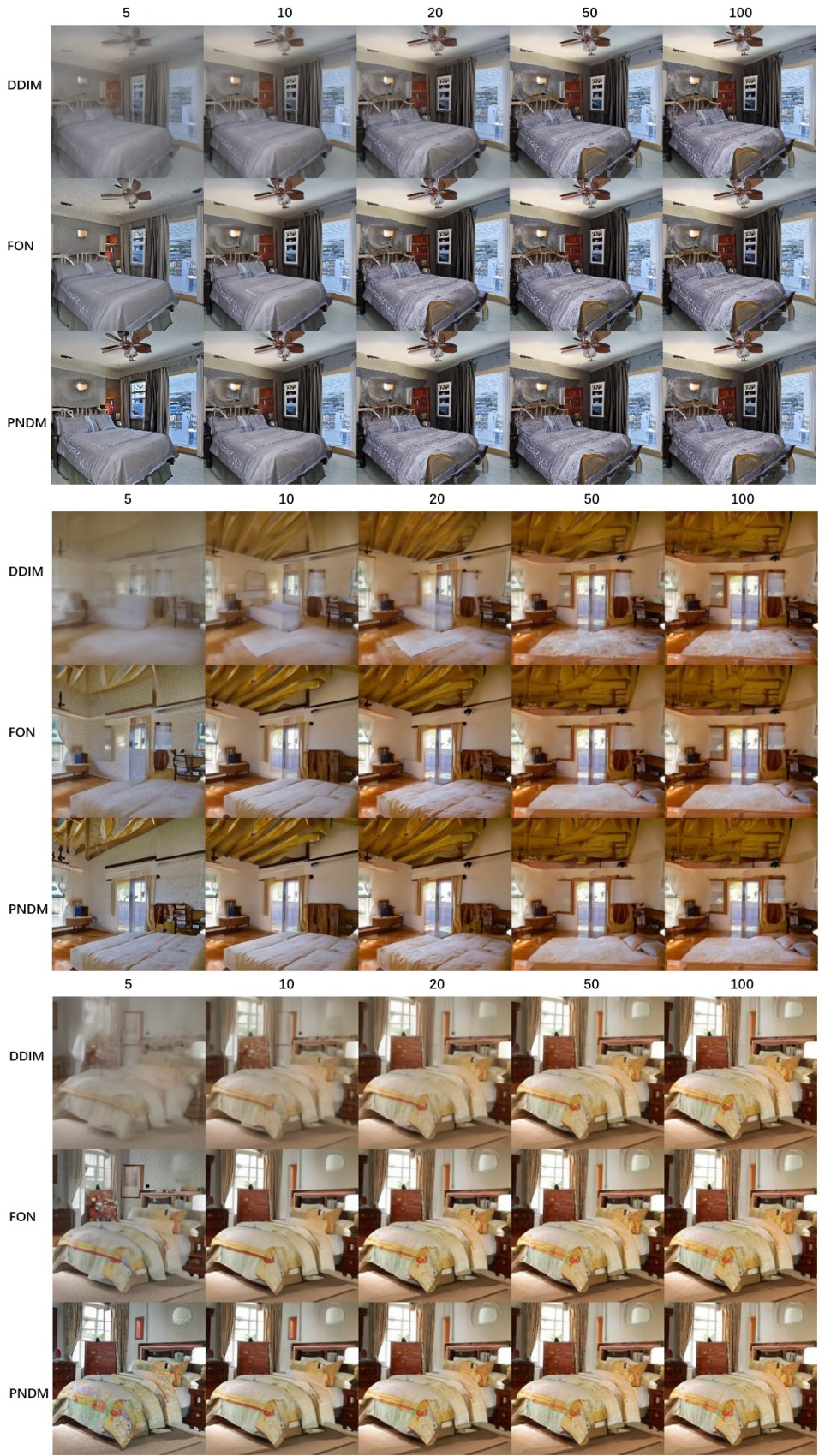

Figure 12: 5, 10, 20, 50, 100-steps generated results using DDIMs, classical numerical methods and PNDMs on LSUN-bedroom.

## A.11 MORE VISUALIZATION RESULTS

Here, we put more visualization results similar to Figure 4 in Figure 13.

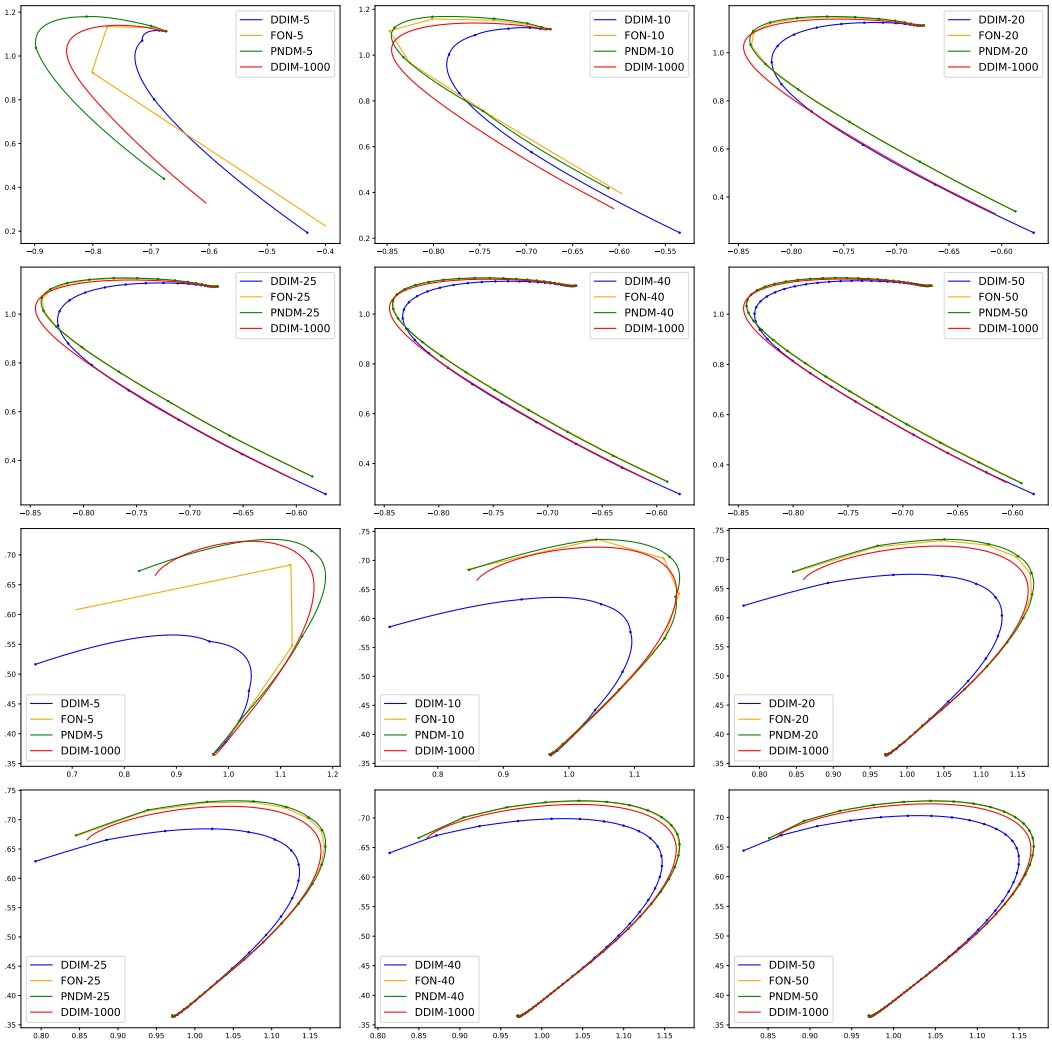

Figure 13: Visualization results under 5, 10, 20, 25, 40 and 50 steps.

## A.12 FID RESULT WITH STANDARD DEVIATION

Here, we report the mean and standard deviation of FID results, tested over four sampling runs.

| dataset | FID\step model | 10 | 20 | 50 | 100 | 250 | 1000 |
|---------|------|------|------|------|------|------|------|
| | DDIM* | 18.50±.06 | 10.86±.08 | 6.95±.04 | 5.49±.06 | 4.52±.02 | 4.02±.04 |
| Cifar10 | FON | 13.00±.11 | 7.33±.06 | 5.24±.05 | 4.64±.04 | 4.12±.03 | 3.73±.03 |
| (linear) | S-PNDM | 11.58±.10 | 7.53±.07 | 5.15±.05 | 4.34±.03 | 3.93±.02 | 3.83±.03 |
| | F-PNDM | 6.12±.07 | 5.04±.07 | 4.01±.02 | 3.75±.04 | **3.67±.03** | 3.78±.04 |

Table 7: Image generation measured in FID on Cifar10. DDIM* means a kind of pseudo numerical method and also a retest of DDIM.

