# OpenReview forum: "Pseudo Numerical Methods for Diffusion Models on Manifolds"
_ICLR.cc/2022/Conference — ICLR 2022 Poster_

### Official Review · Reviewer_c9bY · 2021-11-01

**Correctness:** 3
**Technical Novelty And Significance:** 3
**Empirical Novelty And Significance:** 4
**Recommendation:** 5
**Confidence:** 3

**Main Review:**

## Clarity.
Overall, I believe that the clarity of the submission should be enhanced.

First, the introduction can be improved to better stress the motivation of the submission. If I understand correctly, this work builds on Probability Flows (Song et al., 2020), which leverage the existence of a deterministic process whose trajectories have the same densities as the original diffusion process. This deterministic process satisfies an ODE that depends on the score original drift and diffusion terms but also on the score function. Consequently classical numerical ODE solvers (e.g. RK) can be leveraged to sample data from the probabilistic model. Authors then state that results obtained via this approach are subpar and suggest that this is due to the solvers tendency to sample data far from the data manifold.

1/ Why is this true? Would be necessary to give some intuition and to refer to a theoretical analysis

2/ What is the precise problem? Is the issue that the model oversample the data distribution's tail (i.e. fails to fit the distribution properly) or that the numerical methods fail (in what sense?) in these areas as the score is undefined (or hard to estimate)? SMLD (Song and Ermon, 2020)'s motivation to inject noise is built on the latter.

The authors then provide "pseudo numerical methods for diffusion models (PNDMs)" which produces trajectories that "iterate data on the high-density region of the data", hence tackling the aforementioned issue.

Also, perhaps this is question of taste, but I believe that the Background Section (and most of the paper)'s clarity could be greatly improved by taking the perspective of Song et al. (2020b), that is, a continuous diffusion process (forward / perturbating data) and the associated reverse diffusion process (generating data).

Additionally, Section 3.1 is challenging to follow. Would perhaps be better to put less equations but spend more time explaining why and how they matter.

## Strengths
First, the proposed method is conceptually simple and showcase previously proposed methods (DDIMs) as a special case where the update is a (first-order) Euler step.

Then, the submission shows strong empirical results on common datasets like CelebA, with faster convergences or significantly better FID (for the same number of steps) yielding SOTA. Authors report a ~x20 speed-up wrt DDIM, but this in number of steps and fourth order methods trade-off convergence speed for computational cost. Figure 3 suggests that the speed-up is around x15 for CIFAR which is still significant (although it would ideal to report the runtime directly in Table 2).

Finally, Figure 4 is quite nice as it empirically illustrate the proposed method ability to sample trajectories that like closer to the data manifold, which was the original motivation.

## Weaknesses
I think that the main weakness is the writing.
Although I believe that Section 4.3 would deserve a deeper empirical analysis as it is directly investigating the core motivation of this submission.

## Relation to prior work
- Perhaps worth citing Sohl-Dickstein et al., 2015 in Section 1?
- Citations for Runge-Kutta and the Linear Multi-Step methods appear to be missing.
- It is not entirely clear to me what method is meant by Probability Flows (Song et al., 2020b), is it with Variance Exploding SDE (SMLD) or Variance Exploding SDE (SMLD) (cf Table 1 from that paper).


## Additional feedback.
- "However, classical numerical methods (Sauer, 2017) have problems when they are applied to DDPMs." -> What class of numerical methods? The Euler and RK methods are ODE solvers, although extensions exist for SDEs. It is unclear how they can be applied to DDPMs. Or is it implicitly implied that they are used for the the corresponding deterministic process (probability flow)?
- "to iterate our data on the high-density region" -> would suggest to reformulate
- Eq 3: $\epsilon_\theta$ is not defined. Would advise to do so so for the paper to be self-contained, especially as $\epsilon_\theta$ is used through the entire paper.
- Table 2: Error bars / confidence intervals are missing. Bold is not defined. As methods have different computational cost per step would be very useful to additionally show this metric.
- Figure 3: Time has no unit.
- Figure 4: Axis names are missing.


**Summary Of The Paper:**

Highlighting the high computational complexity for sampling from Denoising Diffusion Probabilistic Models (DDPMs) (e.g. wrt GANs), authors build on the connection between diffusion processes and ODEs to propose efficient (pseudo-)numerical methods so as to sample data from the data manifold. The main idea is to combine the discrete update proposed in DDIMs, with a fourth-order gradient estimation given by the Runge-Kutta or linear multi-step methods. The motivation being that such gradient estimator should yield trajectories that stay closer to the data manifold. They empirically assess their method(s) on CIFAR10 and CelebA in terms of sample quality (measured by FID), and show that they get a ~x20 speed-up wrt DDIMs or a significant improvement in FID with the same number of steps.


**Summary Of The Review:**

I personally find this submission interesting and significant, yet believe that clarity needs to be improved to enable readers take the most of the paper's insights.

---

> ### Author Response · Authors · 2021-11-18
> **Thank you for questions and feedback (1)**
>
> We thank the reviewer for the thoughtful feedback. We are delighted that the reviewer thinks our submission interesting and significant. We believe that the reviewer raised many valuable questions and we believe those questions have improved our work very much. We highlight the main modifications in the new version of the paper.
>
> Below we address specific questions and comments:
>
> 1. **["It is not entirely clear ..."]** First of all, we need to explain the relationship of DDPMs, DDIMs, VP-SDEs and PFs. VP-SDEs and PFs come from Song et al. (2020b) (PFs are defined in Section 4.3 there). The difference between them is VP-SDEs have random items, while PFs do not. This paper shows that these two methods have the same marginal probability density. DDIMs also remove the random items of DDPMs. Therefore, in our paper, we put DDPMs and VP-SDEs together and DDIMs and PFs together. And we add more explanation why we choose to solve the case that the reverse process contains no random items at the beginning of Section 3.1. Therefore, we put more attention on DDIMs and PFs.
>
> 2. **["1/ Why is ..."] & ["2/ What is ..."]** Our main target is to find better numerical methods to solve Equation (10). Numerical methods used in prior works face two main problems the score is meaningful only in a certain area (The first problem in Section 3.2) and the numerical methods fail (The second problem in Section 3.2). The reason why the numerical methods fail is that Equation (10) is unbound near zero, which does not satisfy the conditions of classical numerical methods. In Section 3.3, we show why our new pseudo numerical methods can those two problems (in a new paragraph):
>    1. For the first problem, our new transfer parts do not introduce new errors. This property also means that it keeps the results at the next step on the target manifold because generating samples away is a kind of error.
>    2. For the second problem, we know that the prediction of $\epsilon_\theta$ is more and more precise in the reverse process in the above subsection. And our new transfer part can generate precise results according to the precise prediction of $\epsilon_\theta$. Therefore, our generation results are more and more precise using pseudo numerical methods, while classical numerical methods can introduce obvious error at the last several steps.
>
> 3. **["Also, perhaps this ..."]**  We make a simple introduction of the reverse diffusion process of Song et al. (2020b), namely, the reverse diffusion process of VP-SDEs and PFs in Section 2.2, and add a new explanation of the relationship between VP-SDEs and PFs. PFs can also be treated as an acceleration of DDIMs, while VP-SDEs are an acceleration of DDPMs.
>
> 4. **["Additionally, Section 3.1 ..."]** We add more explanations for the three main equations in Section 3.1, we try to show why we compute these equations and what we do to compute these equations. Here, we first remove the random items and then change DDIM to a differential form. Finally, we can compute the corresponding ODE of DDIM.
>
> 5. **["I believe that Section 4.3 ..."]** We add a new toy example to support Section 4.3 and get similar results with the real cases. For the lack of space, we put the details of this toy example in Appendix A.8.
>
> 6. **["Perhaps worth ..."] & ["Citations for ..."]** Thanks for your advice! We add Sohl-Dickstein et al. (2015) at the beginning of the introduction and put the citations for RK method and linear multi-step method in the introduction and Section 2.3.
>
> 7. **["However, classical numerical ..."]**  We make this statement more clear in the new version of the paper and replace this sentence with "this direct connection between DDPMs and numerical methods (e.g., forward Euler method, linear multi-step method and Runge-Kutta method (Timothy, 2017)) has weaknesses in both speed and effect." to show what numerical methods we are talking. And Section 3.1 introduces how to build the connection of DDPMs and these numerical methods in detail.
>
> 8. **["Eq 3: $\epsilon_\theta$ is not ..."]** In Section 2.1, we add the definition of $\epsilon_\theta$ , $\alpha_t$ and more explanations of some equations to make sure our background is self-contained. For example, we show how to choose $\beta_\theta$ in Section 3.1.

---

> > ### Author Response · Authors · 2021-11-22
> > **Thank you for questions and feedback (2)**
> >
> > 9. **["Table 2: Error bars ..."] & ["Figure 3: ..."]** We test the computation cost under different methods and add them in Table 2, and use the computation cost per step of DDIMs as the unit of Figure 3. We find that the computation costs using the same pretrained models are similar.
> >
> >    |                  | DDIM  | FON   | S-PNDM | F-PNDM |
> >    | ---------------- | ----- | ----- | ------ | ------ |
> >    | Cifar10 (linear) | 0.337 | 0.390 | 0.344  | 0.391  |
> >    | Cifar10 (cosine) | 0.505 |       | 0.517  | 0.595  |
> >    | Celeba (linear)  | 1.237 | 1.431 | 1.258  | 0.433  |
> >
> >    Here, we use the 50-step, 512 batch size experiment on an RTX-3090 to test the computational cost and the column time is the average computational cost per step in seconds.
> >
> >    And we are testing the error bars / confidence intervals, which need a little more time. For now, we have completed the standard deviation test on Cifar10 (linear) using the pretrained model from Ho et al. (2020) and put the results in Appendix A.12. The results are as follows:
> >
> >    |           | 10            | 20            | 50           | 100          | 250          | 1000         |
> >    | --------- | ------------- | ------------- | ------------ | ------------ | ------------ | ------------ |
> >    | DDIM*     | 18.50$\pm$.06 | 10.86$\pm$.08 | 6.95$\pm$.04 | 5.49$\pm$.06 | 4.52$\pm$.02 | 4.02$\pm$.04 |
> >    | FON       | 13.00$\pm$.11 | 7.33$\pm$.06 | 5.24$\pm$.05 | 4.64$\pm$.04 | 4.12$\pm$.03 | 3.73$\pm$.03 |
> >    | S-PNDM    | 11.58$\pm$.10 | 7.53$\pm$.07 | 5.15$\pm$.05 | 4.34$\pm$.03 | 3.93$\pm$.02 | 3.83$\pm$.03 |
> >    | F-PNDM    | 6.12$\pm$.07 | 5.04$\pm$.07 | 4.01$\pm$.02 | 3.75$\pm$.04 | 3.67$\pm$.03 | 3.78$\pm$.04 |
> >
> > 10.  **["to iterate ..."] & ["Figure 4: ..."]** We also fix the problems in Figure 4 and sentence "to iterate our data on the high-density region".
> >
> > Thanks again for your comments on our paper. If our replies feel satisfactory, we would like to kindly ask the reviewer to consider raising the score accordingly. At the same time, we are happy to discuss further.

---

> > ### Comment · Reviewer_c9bY · 2021-11-28
> > **Re: Thank you for questions and feedback**
> >
> > I thank the authors for taking the time to answer my questions, giving additional empirical results and for updating the submission accordingly.
> > In particular, I feel that the motivation and the the relationship with previous methods have been improved.
> >
> > Yet, I believe that the exposition of the approach, and the overall writing, still needs significant improvement. There are quite a few statements that are not entirely rigorous, e.g. `This is Probability Flows (PFs), which remove the random item of VP-SDEs`, or `precise` not being defined. Additionally, clarity can and should be improved. For instance `S-PNDM` and `F-PNDM` could be renamed respectively `PNDM(2)` and `PNDM(4)` as they are based respectively on second and fourth order methods, would also rename `FON` by `RK(4)`. What is more, I would again suggest taking the notations and perspective of (score-based) reverse diffusion processes (Song et al., 2020b), which would likely significantly simplify equations and enhance the flow (e.g. parametrising the score network $s_\theta$ instead of $\epsilon_\theta$). I would suggest putting less equations, and focusing on the ones that are key for the narrative of the paper, with an emphasis on giving intuition to the reader. Furthermore, it is still unclear how Table 2 relates to Table 1 from Song et al. (2020b).
> >
> > I am sympathetic to the authors' effort to improve the submission, and do believe that the idea is interesting and can be practically useful, but I am not convinced that the submission reaches (yet) the expected quality for ICLR. I may update my score depending on the discussion with other reviewers.

---

> > > ### Author Response · Authors · 2021-11-28
> > > **Re: Re: Thank you for questions and feedback**
> > >
> > > Thanks for the reviewer's additional feedback. Below we address specific questions and comments:
> > >
> > > 1. **["`S-PNDM` and `F-PNDM` could ..."]** Thanks for your advice! We will change the name of S-PNDM and F-PNDM in the new version of the paper. However, FON is not RK(4). Similar to F-PNDM, it uses Runge-Kutta method at the beginning and linear multi-step in the following. So we call this method the fourth-order numerical method (FON) instead of RK or LMS. And we will add this explanation to the paper.
> > > 2. **["This is Probability ..."]** & **[" I would again suggest ..."]** We will add more introduction about Song et al. (2020b) at the beginning of Section 2.2. What's more, we will add a more detailed explanation for the relationship between PFs and VP-SDEs and make a more rigorous statement. However, we strongly recommend the reviewer and readers to read the detailed analyses of this relationship in Appendix D of Song et al. (2020b).
> > > 3. **["which would likely significantly simplify equations ..."]** In Section 3.1, we give a self-contained and simple formula transformation to get the corresponding ODE of DDIM, which is a contribution of our work. By contrast, Song et al. (2020b) use a big theorem to get a similar result. Therefore, we do not use some perspectives in this paper. However, we will recheck our notations to find whether we can use notations from this paper to simplify our equations and thanks for your advice.
> > > 4. **["Furthermore, it is still unclear ..."]** Our baseline is the result using predictor probability flow (PF), sampler P1000 and method VP-SDE in Table 1 from Song et al. (2020b), which is the most similar case with our experiments. Because all experiments in our paper use at most 1000 steps and PC sampler is a trick, which can be used in both works. However, this paper does not provide the results under different numbers of steps, so we retest them by ourselves in Table 2. And we will add an explanation for this choice in the new version of the paper.
> > >
> > > We will try our best to make the paper both more clear and rigorous. If the reviewer has more advice for the paper's clarity, we are happy to change all of them in the final version of the paper to make sure our paper satisfies the expected quality.

---

### Official Review · Reviewer_urhS · 2021-11-02

**Correctness:** 4
**Technical Novelty And Significance:** 3
**Empirical Novelty And Significance:** 4
**Recommendation:** 8
**Confidence:** 3

**Main Review:**

The paper introduces a class of pseudo numerical methods for DDPM, this is based on a previous work that already establishes the relationship between DDPMs and a certain class of differential equations on manifolds. The pseudo numerical methods separate the gradient and transfer step of classical numerical algorithms, and for each part choose the best of both worlds. With this they are able to provide faster converengence and efficient update steps because gradients do not need to be recomputed for every step. While I would not guarantee that methods do not exist already in classical optimization, I have not seen them in any related application.

The paper is overall well written, and the contribution and main ideas are explained clearly. Starting at Section 3.3 (to 4) the derivations become slightly confusing, and it takes a lot of referencing back to find all the variable names again. A legend for the variables and a bit more redundant explanation would probably help here. The same goes for the derivations in the appendix which I could not follow completely.

Experiments are done on four datasets with different resolutions using pretrained models for the manifold description. The results show that the introduce pseudo numerical methods converge much faster and provide better results in less iterations than the previous DDIM method and classical numerical methods. The qualitative examples in the appendix look good, but some of them are too smooth to compete with general sota generators (I am not sure if the smoothness here is related to the pretrained models that were used?).

Minor comments:
- The authors claim that their implementation is in the supplementary. I was not able to find any supplementary (except the appendix) but I am honestly not sure if this is a failure of me using OpenReview... The implementation should definitely be published in the end though.
- I think it is bad practice to move the related work section in the appendix.
- The large figures in the appendix are very hard to understand because the subfigures are not separately titled.
- There are some grammatical errors throughout the text which should be proofread again.

**Summary Of The Paper:**

The paper proposes a new efficient method for denoising diffusion probabilistic models (DDPM) (generative models that optimize for the closest solution on a manifold) based on the observation that this can be seen a solving a set of differential equations on a manifold. This allows efficient pseudo numerical methods to be applied here which have many advantageous properties over classical optimization methods, including less optimization steps and guaranteed manifold solutions due to separating the gradient part from the transfer part in the optimization. Results are shown on four datasets comparing to two reasonable but not sota baselines.

**Summary Of The Review:**

The idea of separating the gradient and transfer part is (to the best of my knowledge, I am not an expert though) novel, and I can see many applications besides the ones proposed in this paper. The shown results might not be exactly state-of-the-art in terms of generating images, but show clear advantages over traditional numerical methods. Therefore I recommend accept.

---

> ### Author Response · Authors · 2021-11-18
> **Thank you for questions and feedback**
>
> We thank the reviewer for the very positive feedback. We are delighted that the reviewer thinks that many applications besides the ones proposed in our paper. We believe that the reviewer raised many useful questions and we believe those questions have improved our work very much. We highlight the main modifications in the new version of the paper.
>
> Below we address specific questions and comments:
>
> 1. **[About Section 3.3 (to 4)]** Section 3 introduces our main theoretical results and we are trying our best to make it more clear and read-friendly. We add more explanations to show why we use Equation (11) as the transfer part and add a new paragraph to explain the benefits of this choice. In this new paragraph, we show that our new methods successfully solve two main problems mentioned in Section 3.2. We also offer two main properties as Property 3.1 and Property 3.2, to help the reviewer and readers better understand our main points. What's more, we also add more explanations of many variables. I hope these explanations can help the reviewer and readers.
> 1. **[About generation examples in Appendix]** This is a problem from the original DDPM, and we want to accelerate the reverse process and use pretrained DDPMs. Therefore, it is hard to solve this problem in this paper. Maybe we will explore this area in future works.
> 1. **[About supplementary material]** We put our implementation at [Supplementary Material](https://openreview.net/attachment?id=PlKWVd2yBkY&name=supplementary_material) and hope this can help the reviewer and readers.
> 1. **[About related work]** Due to the limited space of the main paper, I put the most important related works in the introduction and put more related works in Appendix A.1. And hope this solution is helpful for the reviewer and readers.
> 1. **[About the captions of figures]** We marked large figures in the Appendix to make the meaning of each picture clearer.
> 1. **[About grammatical error]** Sorry for that. We have corrected some errors in the new version of the paper. We highlight all modifications in the new version of the paper.
>
> We hope that we were able to address all the reviewer's questions and concerns. And we are delighted to further discuss.

---

### Official Review · Reviewer_jhKu · 2021-11-04

**Correctness:** 3
**Technical Novelty And Significance:** 3
**Empirical Novelty And Significance:** 3
**Recommendation:** 5
**Confidence:** 2

**Main Review:**

The beginning of the article (Section 1 and Section 2) is well written and presents the motivation behind the introduction of these pseudo numerical methods in a very didactic way. However, Section 3 needs a lot of polishing. These are some comments and suggestions:

1. The contribution of this work is based on the assumption that $\sigma_t = 0$. The conditions required to satisfy this assumption deserve to be commented.


2. The mention of the manifolds is very implicit and requires to be clarified: what is the definition of the manifold defined at the beginning of Section 3.2?

3. The paragraph at the beginning of Section 3.3, supposed to give the intuition behind the introduction of the transfer part defined in (11), is not clear and should be reformulated. For example, what do you mean by ‘We find that Equation (9) has the property that if ϵ is the precise noise in $x_{t}$, then the result of $x_{t−\delta}$ is also precise, no matter how big $\delta$ is’?

4. In Algorithm 2, PLMS and PRK were not defined


5. It would be preferable to put the theoretical results of Section 3.6 in a clearly stated theorem. This would highlight the theoretical contributions of this work.








**Summary Of The Paper:**

This article studies pseudo numerical methods that improve and accelerate upon exiting numerical methods for Denoising Diffusion Probabilistic Models (DDPMs). The crux of this work is the observation that existing work on this topic does not take into consideration the structure of the high-density region of the data.

**Summary Of The Review:**

The novelty of the methodology presented in this paper qualifies this work to be accepted to the conference conditionally to improve the clarity of Section 3.

---

> ### Author Response · Authors · 2021-11-18
> **Thank you for questions and feedback**
>
> We thank the reviewer for the thoughtful feedback. We are delighted that the reviewer agrees with us that our pseudo numerical methods are novel. We think that the reviewer raised many useful questions and we believe those questions have improved our work very much. We highlight the main modifications in the new version of the paper.
>
> Below we address specific questions and comments:
>
> 1. **["1. The contribution of ..."]** Thanks for your advice. $\sigma_t=0$ means that we can find a corresponding ODE without random items while solving differential equations with or without random items are different. Theoretically, there are more choices to solve differential equations without random items. Empirically, Song et al. (2020a) show that DDIMs have a better acceleration effect when the number of total steps is relatively small. We add these explanations at the beginning of Section 3.1 in the new version of the paper.
> 2. **["2. The mention of ..."]** We add a clear definition of our target manifolds. These manifolds are the high-density region of the data $x_t$ of DDPMs, which is defined by $x_t(x_0, \epsilon)= \sqrt{\bar{\alpha}_t}x_0 + \sqrt{1-\bar{\alpha}_t}\epsilon$, $\epsilon\sim \mathcal{N}(0,1)$. We put this definition in the introduction and at the beginning of Section 3.2.
> 3. **["3. The paragraph at ..."]** To explain the choice of our new transfer part, at the beginning of Section 3.3, we first use Property 3.1 to try to explain why we choose Equation (11) as the transfer part. After Equation (11), we  explain the benefits of this choice and how this choice solves the problems in classical numerical methods: (We make these explanations as a new paragraph in Section 3.3 in the new version of the paper)
>    1. our new transfer parts do not introduce new errors. This property also means that it keeps the results at the next step on the target manifold because generating samples away is a kind of error.
>    2. we know that the prediction of $\epsilon_\theta$ is more and more precise in the reverse process in the above subsection. And our new transfer part can generate precise results according to the precise prediction of $\epsilon_\theta$. Therefore, our generation results are more and more precise using pseudo numerical methods, while classical numerical methods can introduce obvious error at the last several steps.
>
> 4. **["4. In Algorithm 2, ..."]** We put the definition of PRK and PLMS below Equation (13) and Algorithm 2.
> 5. **["3. The paragraph at ..."] & ["5. It would be ..."]** Thanks for your advice! In Section 3.3 and 3.6, we clearly state two properties. Property 3.1 reformulates, "We find that Equation (9) has the property that if $\epsilon$ is the precise noise in $x_t$, then the result of $x_{t-\delta}$ is also precise, no matter how big $\delta$ is.". And we put the proof of Property 3.1 in Appendix A.5. What's more, we also add "our new methods PNDMs are second-order convergent while DDIMs are first-order convergent." in the introduction.
>
> Thanks again for your comments on our paper. If our replies feel satisfactory, we would like to kindly ask the reviewer to consider raising the score accordingly. At the same time, we are happy to discuss further.

---

> > ### Author Response · Authors · 2021-12-04
> > **Looking foward to your feedback**
> >
> > Dear reviewer jhKu,
> >
> > We hope you've had a chance to read our response and the revised paper. We would really appreciate a reply before the end of the discussion period about whether we have addressed your concerns or if any additional concerns remain. We are happy to address any remaining concerns and eagerly look for your feedback on the revised paper.
> >
> > The authors of "Pseudo Numerical Methods for Diffusion Models on Manifolds."

---

### Official Review · Reviewer_HRpL · 2021-11-07

**Correctness:** 3
**Technical Novelty And Significance:** 2
**Empirical Novelty And Significance:** 3
**Recommendation:** 6
**Confidence:** 2

**Main Review:**

I thank the authors for this submission. I believe there is value in this work both from a theoretical and practical perspective. In general, I am willing to accept the paper. I have two main suggestions:

1. Please double-check the writing and notations. Some sentences are hard to understand and equations contain errors, e.g. Eq. 2, hat{alpha}_t, wrong index.

2. As DDPMs are relatively new, I was expecting a bit more elaborate introduction.

3. I was honestly quite disappointed with the presentation at times. Some equations are stated without any intuition. If you are short of space, I would rather move some of the experimental parts to a supplementary.

4. Sec. 4.3 and in general, I would also work with toy problems and easily visualizable data. This will provide additional insight for the reader.

**Summary Of The Paper:**

This paper introduces a framework to treat Denoising Diffusion Probabilistic Models (DDPMs) as solving differential equations on manifolds. This goal is faster sampling without significant loss of quality.

**Summary Of The Review:**

I am not very familiar with DDPMs but as far as I can tell, the numerical technique introduced makes sense and leads to improved results. I, therefore, recommend acceptance.

---

> ### Author Response · Authors · 2021-11-18
> **Thank you for questions and feedback**
>
> We thank the reviewer for the positive feedback. We are delighted that the reviewer agrees that there is value in our work both from a theoretical and practical perspective. We think that the reviewer raised many valuable questions and we believe those questions have improved our work very much.
>
> Below we address specific questions and comments:
>
> 1. **[About the writing and notations]** Sorry for that. We have corrected some errors in the new version of the paper. And we highlight modifications in the new version of the paper.
> 2. **[About more elaborate introduction]** We add more definitions and more explanations of some equations in Section 2 to make this section more self-contained and give readers a better reading experience. But Section 2 still lacks some details of original DDPMs. Therefore, we recommend readers read three main original papers, including Song et al. (2020a), Song et al. (2020b) and Ho et al. (2020). To be more specific, we show how to choose $\beta_\theta$ in Section 3.1 and the relationship between Equation (4) and (5). The main difference between them is PFs have no random items.
> 3. **[About more intuitions and benefits of equations]** We have added more explanations for Equation (3), (4), (5), (8), (9) and (11) in the new version of the paper to improve the presentation and help readers understand our methods better. For Equation (8), we explain why we choose $\sigma_t=0$. The main idea is that we can turn the reverse process of diffusion models into solving an ODE without random items. For Equation (11), we first give more intuition behind our choice and then use a new paragraph to show that this choice solves the problems mentioned in Section 3.2.
> 4. **[About a toy example for Section 4.3]** We add a toy example for Section 4.3 and get similar results with the real cases. And we find that the results are really similar to the real cases. Due to the limitation of the main paper, we put this example in Appendix A.8.
>
> Thanks again for your comments on our paper. If our replies feel satisfactory, we would like to kindly ask the reviewer to consider raising the score accordingly. At the same time, we are happy to discuss further.

---

### Author Response · Authors · 2021-11-25
**A summary of the main changes**

Dear reviewers,

Here we add a summary of the main changes we made to the article based on the comments and suggestions of the reviewers:

1. In the introduction, we add new sentences to point out the two main problems of classical numerical methods. And in this paper, we design new numerical methods (PNDMs) to solve these problems and PNDMs achieve remarkable improvement.
2. In Section 2, we correct some errors and add new sentences to make the background more self-contained.
3. In Section 3.1, we add a new paragraph to explain why we choose $\sigma_t=0$.
4. In Section 3.2, we add a more precise definition of the main manifolds mentioned in this paper.
5. In Section 3.3, we first provide Property 3.1 to show the intuition of Equation (11) and add the proof of Property 3.1 in Appendix A.5. Then, we add a new paragraph to show how this choice can solve the two problems of classical numerical methods.
6. In Section 3.6, we make our main theoretical contribution as Property 3.2.
7. In Section 4.3, we add a new toy example to support our analyses and put this toy example in Appendix A.8.
8. In Table 2, we add a computational cost test and put the error bars results in Appendix A.12.

In addition, we would like to kindly remind you that the final stage of discussion ends in 4 days. It would be great if you could have a look at our replies and let us know if our responses are satisfactory or if there are any further follow-up questions. And we are happy to discuss this further at any time.

The authors of "Pseudo Numerical Methods for Diffusion Models on Manifolds."

---

### Public Comment · ~Luping_Liu2 · 2022-02-10
**Code release**

Our official PyTorch implementation and checkpoints are released at GitHub: [https://github.com/luping-liu/PNDM](https://github.com/luping-liu/PNDM).

This code is not only the official implementation for PNDM, but also a generic framework for DDIM-like models including:
- [Pseudo Numerical Methods for Diffusion Models on Manifolds (PNDM)](https://openreview.net/forum?id=PlKWVd2yBkY)
- [Denoising Diffusion Implicit Models (DDIM)](https://arxiv.org/abs/2010.02502)
- [Score-Based Generative Modeling through Stochastic Differential Equations (PF)](https://arxiv.org/abs/2011.13456)
- [Improved Denoising Diffusion Probabilistic Models (iDDPM)](https://arxiv.org/abs/2102.09672)

This code contains three main objects including method, schedule and model. The following table shows the options
supported by this code and the role of each object：

| Object   | Option                        | Role                                          |
|----------|-------------------------------|-----------------------------------------------|
| method   | DDIM, S-PNDM, F-PNDM, FON, PF | the numerical method used to generate samples |
| schedule | linear, quad, cosine          | the schedule of adding noise to images        |
| model    | DDIM, iDDPM, PF, PF_deep      | the neural network used to fit noise          |

---

### Decision · Program_Chairs · 2022-01-20

**Decision:**

Accept (Poster)

**Comment:**

This paper presents a new DDPM model based on solving differential equations on a manifold.  The resulting numerics appear to be favorable, with faster performance than past models.

Most of the reviews thought the main result was of interest and were impressed with the performance.  Reviewer c9bY points out some challenging issues and analytical questions that remain unanswered in the text; they also have some simpler textual revisions that seem less important.

In general, this paper has the misfortune of receiving reviews whose confidence appears to be low.  While partially this is a byproduct of the noisy machine learning review system, the difficulty of the text itself is substantial and made the paper less than approachable; the authors are encouraged to continue to revise their text based on feedback from as many readers as possible.  That said, the authors were quite responsive to reviewer comments during the rebuttal phase, which significantly improved the text.

Overall this is a borderline case, and the AC also had some difficulty following details of this technically dense paper.  Given the positive *technical* assessments of the work and at least one reviewer defending the paper's clarity, the AC is willing to give this paper the benefit of the doubt.